# GSK3A promotes human adenovirus replication and phosphorylates viral L4-22K protein

Ying Lin[1,2,3,4,5,*] , Yun Zhu[2,3,4,5,*] , Ling Jing[1,2,3,4,5] , Yongjun Chen[1,6], Xia Xiao[1,6], Xiaobo Lei[1,6] , Zhengde Xie[2,3,4,5]

Human adenovirus B7 (HAdV-B7) is a significant respiratory pathogen in children, associated with substantial morbidity and mortality. Despite its clinical importance, the molecular mechanisms of HAdV-B7–host interaction remain poorly elucidated. In this study, we performed a high-throughput gain-of-function cDNA library screening and identified glycogen synthase kinase 3α (GSK3A) as a key proviral factor facilitating HAdV-B7 replication. The overexpression of GSK3A enhanced viral replication, whereas knockdown or knockout inhibited it. Furthermore, the kinase-active S21A mutant significantly augmented viral replication, whereas the kinase-inactive Y279A and K148A mutants of GSK3A failed to support it, highlighting the importance of its kinase activity on HAdV-B7 replication. Notably, phosphoproteomic and co-immunoprecipitation assays (co-IP) revealed that GSK3A phosphorylates viral L4-22K protein at S78 and S81 residues in a partially kinase-dependent manner. Using structure modeling, protein–protein docking, and truncation assays, we mapped the interaction between GSK3A's kinase domain and the 92–168 aa region of the viral L4-22K protein. In addition, GSK3A acts as a broad-spectrum proviral factor for respiratory HAdV, particularly for Species B, corresponding to a high similarity in L4-22K sequences. As a result, we identified GSK3A as a crucial proviral host factor for HAdV replication and provided a promising avenue for targeting GSK3A in the development of antiviral therapies against HAdV infections.

## Introduction

Human adenovirus (HAdV) is a significant pathogen that causes acute respiratory infection, particularly in children, and is associated with considerable morbidity and mortality (Li et al, 2021b; Pscheidt et al, 2021; Khan et al, 2022; Liu et al, 2023; Abdullah et al,

2024). HAdVs, belonging to the genus *Mastadenovirus* in the family *Adenoviridae*, are nonenveloped viruses with a double-stranded DNA genome, encoding more than 40 viral proteins. As of the latest classification update in March 2024, 116 types have been identified and categorized into seven species (A-G) based on their antigenic and genomic characteristics (http://hadvwg.gmu.edu/). Among them, human adenovirus B7 (HAdV-B7) poses a significant health threat, particularly to children, because of its association with higher risks of severe pneumonia, toxic encephalopathy, acute respiratory dysfunction syndrome, and even death cases (Tang et al, 2011; Liu et al, 2015; Hai et al, 2016; Fu et al, 2019; Xie et al, 2019; Zou et al, 2021). Epidemiological data have indicated that certain HAdV types, particularly HAdV-B7, are more prevalent in children (Liu et al, 2015, 2023; Xu et al, 2018; Li et al, 2021a; Chen et al, 2022). Currently, there are no effective vaccines or antiviral treatments for HAdV-B7, underscoring the need for better insights into the viral replication process and host factors that affect viral infection.

Viruses rely on host cellular machinery to complete their replication cycle, making host proteins essential for understanding viral pathogenesis. Identifying host proteins that regulate viral replication can provide insights into viral pathogenesis and uncover potential therapeutic targets. Previous studies have reported several host factors that modulate HAdV replication, such as the proviral mind bomb 1 (Mib1), interleukin-8 (IL-8), tank binding kinase 1 (TBK1), ubiquitin-specific peptidase 7 (USP-7) (Ching et al, 2013; Kotha et al, 2015; Chikhalya et al, 2021; Pied et al, 2022). In addition, antiviral factors like tripartite motif-containing protein 21 (TRIM21), TRIM35, interferon-induced protein with tetratricopeptide repeats 3 (IFIT3), and apolipoprotein B mRNA-editing enzyme catalytic polypeptide 3 (APOBEC3) have been identified (McEwan et al, 2013; Chikhalya et al, 2021; Sun et al, 2023). However, most studies have focused on HAdV-C5, with limited investigation into other HAdV species. Notably, only three host proteins regulating HAdV-B7 replication were reported. The proviral high-mobility group box-1 protein (HMGB1) promoted late transcription of

[1]NHC Key Laboratory of System Biology of Pathogens and Christophe Merieux Laboratory, National Institute of Pathogen Biology, Chinese Academy of Medical Sciences and Peking Union Medical College, Beijing, P.R. China   [2]Laboratory of Infection and Virology, Beijing Pediatric Research Institute, Beijing Children's Hospital, Capital Medical University, National Center for Children's Health, Beijing, China   [3]Key Laboratory of Major Diseases in Children, Ministry of Education, National Clinical Research Center for Respiratory Diseases, Beijing, China   [4]Beijing Key Laboratory of Core Technologies for the Prevention and Treatment of Emerging Infectious Diseases in Children, Beijing Research Center for Respiratory Infectious Diseases, Beijing, China   [5]Research Unit of Critical Infection in Children, Chinese Academy of Medical Sciences, 2019RU016, Beijing, China   [6]Key Laboratory of Pathogen Infection Prevention and Control (Peking Union Medical College), Ministry of Education, Beijing, P.R. China

Correspondence: fyleixb@126.com, leixb@ipbcams.ac.cn; xiezhengde@bch.com.cn
*Ying Lin and Yun Zhu contributed equally to this work

HAdV-B7 promoters, facilitated viral replication, and accelerated pulmonary inflammation by activating NF-κB (Watt & Molloy, 1988; Tang et al, 2018). The antiviral B-cell lymphoma 2–associated athanogene 3 (BAG3) interacts with viral structure protein VI (pVI) and stimulates autophagy to limit virus replication (Zhang et al, 2023). In addition, retinoic receptor β (RARβ) restricts HAdV-B7 replication at a late phase, although the underlying mechanism remains unclear (Wang et al, 2018). Despite these insights, there is a gap in research on host factors that modulate HAdV-B7 replication.

High-throughput gain-of-function screening has emerged as a powerful tool in genomics research, offering significant advantages over loss-of-function studies, especially for gene families with mutual compensation functions. Unlike loss-of-function approaches, which may be limited by redundancy and compensation between genes, gain-of-function screening allows for the direct observation of phenotypic changes resulting from the overexpression of specific genes. In this study, we employed a high-throughput gain-of-function screening approach and identified glycogen synthase kinase 3 α (GSK3A) as a key proviral factor for HAdV-B7 replication. Further phosphoproteomic and immunoprecipitation–mass spectrometry co-analysis uncovered that the kinase activity of GSK3A impacted the phosphorylation landscape of specific viral proteins, particularly at the S78/81 sites of L4-22K, during the intermediate late phase of the HAdV-B7's life cycle. Our findings provide a foundation for exploring GSK3A as a potential target for antiviral therapies against HAdV-B7 infections.

# Results

### High-throughput cDNA library screening

To investigate the host factors involved in HAdV-B7 infection, we conducted a high-throughput gain-of-function screen using HeLa cells, as illustrated in Fig 1A. Screening a library of 15,000 genes, we identified 75 proviral host factors that promoted viral infections and 63 antiviral host factors that restricted viral replication. These factors were determined by setting an absolute $Z$-score threshold for positively infected cell counts higher than 3 or lower than –2, as indicated by red and blue dots, respectively (Fig 1B). Gene Ontology (GO) function analysis of the proviral factors revealed that they are predominantly involved in key biological processes, including peptidyl-serine phosphorylation, gland morphogenesis and development, and lymphocyte differentiation (Fig 1C). In addition, these factors also participate in various molecular functions including PKA binding (Fig 1C). Phosphorylation modification of proteins has played an important role in regulating viral protein functions. Previous studies have identified several key adenoviral proteins regulated by host cell kinases during replication, including E1A, DBP, and DNA polymerase (Russell et al, 1989; Dumont & Branton, 1992; Ramachandra et al, 1993). Two essential kinases of the glycogen synthase kinase 3 (GSK3) family raised our interests, including GSK3A and GSK3B. In our immunofluorescence screening data, only GSK3A showed a proviral potential on HAdV-B7 expression, which significantly increased the viral infection rate in HeLa cells (Fig 1D). To further confirm this, we performed a verified

overexpression experiment to assess the adenoviral DBP protein level based on Western blot analysis. The result indicated a dose-dependent increment of the viral DBP protein level after ectopic overexpression of GSK3A (Fig 1E), but not GSK3B (Fig 1F), consistent with results obtained from the initial cDNA screening. In recent years, research has demonstrated that GSK3A, leveraging its kinase activity, participates in regulating the replication of various viruses, positioning it as a focal point in studies of antiviral immune mechanisms and the development of new therapeutic targets (Alfhili et al, 2020). However, its involvement in adenoviral replication remains unreported. In combination with our screening results, we have prioritized our research to investigate how GSK3A regulates HAdV-B7 replication.

### GSK3A promotes the replication of HAdV-B7

To confirm the role of GSK3A in HAdV infection, we infected HeLa cells, which had been down-regulated GSK3A expression using siRNA, with HAdV-B7. The results showed that GSK3A knockdown significantly reduced the viral DBP protein level, as well as the transcript levels of viral *DBP*, *E1A*, *hexon* (Fig 2A). To further validate this effect, two GSK3A-KO cell clones were generated using sgRNA and confirmed by immunoblotting and Sanger sequencing (Fig 2B). The cell lines were named HeLa-GSK3A-KO1 and HeLa-GSK3A-KO2, short for KO1 and KO2, respectively. Moreover, the expression of the viral DBP protein indicated that the depletion of GSK3A resulted in a marked decrease in viral replication in both of the KO1 and KO2 cell (Fig 2C). This down-regulation was mirrored at the transcript level of the viral *E1A*, *DBP*, and *hexon* genes (Fig 2D). In addition, the virus titers of HAdV-B7 were significantly reduced (>1 log) in the absence of GSK3A expression (Fig 2E). Together, these findings demonstrated that GSK3A was essential for efficient replications of HAdV-B7.

In addition, we explored the effect of exogenous GSK3A expression on HAdV-B7 propagation in WT and GSK3A-KO cells. The overexpression of GSK3A enhanced the viral DBP protein level in a dose-dependent manner (Fig 3A). This phenomenon was paralleled by elevated transcription of viral genes, including *DBP*, *E1A*, and *hexon* (Fig 3B). The TCID$_{50}$ assay further confirmed a significant increase in viral titer, rising from 3.77 ± 0.59 TCID$_{50}$/ml in the control group to 4.71 ± 0.31 TCID$_{50}$/ml in cells transfected with 500 ng GSK3A with ~1.25-fold increase (Fig 3C).

To eliminate the influence of the endogenous GSK3A expression, a plasmid encoding GSK3A was transfected into HeLa-GSK3A-KO cells. Moreover, the exogenous expression of GSK3A restored the susceptibility of HeLa-GSK3A-KO cells to HAdV-B7 infection. In detail, DBP protein levels increased in a dose-dependent manner after GSK3A restoration (Fig 3D). In addition, the mRNA levels of viral genes *E1A*, *DBP*, and *hexon* were up-regulated upon GSK3A expression (Fig 3E). Consistently, the virus titer of HAdV-B7 increased, suggesting a specific role of GSK3A in viral replication (Fig 3F).

Correspondingly, we also investigated the effect of exogenous GSK3B on HAdV-B7 production in both WT and HeLa-GSK3A-KO1 cells. As we found, exogenous GSK3B had no impact on viral titers regardless of the presence of GSK3A (Fig 3G). Similarly, viral DBP protein level (Fig 3H) and viral *E1A*, *DBP* transcripts (Fig 3I) remained unchanged upon GSK3B overexpression. Although the over-expression of GSK3B seemed to stimulate viral *hexon* mRNA levels

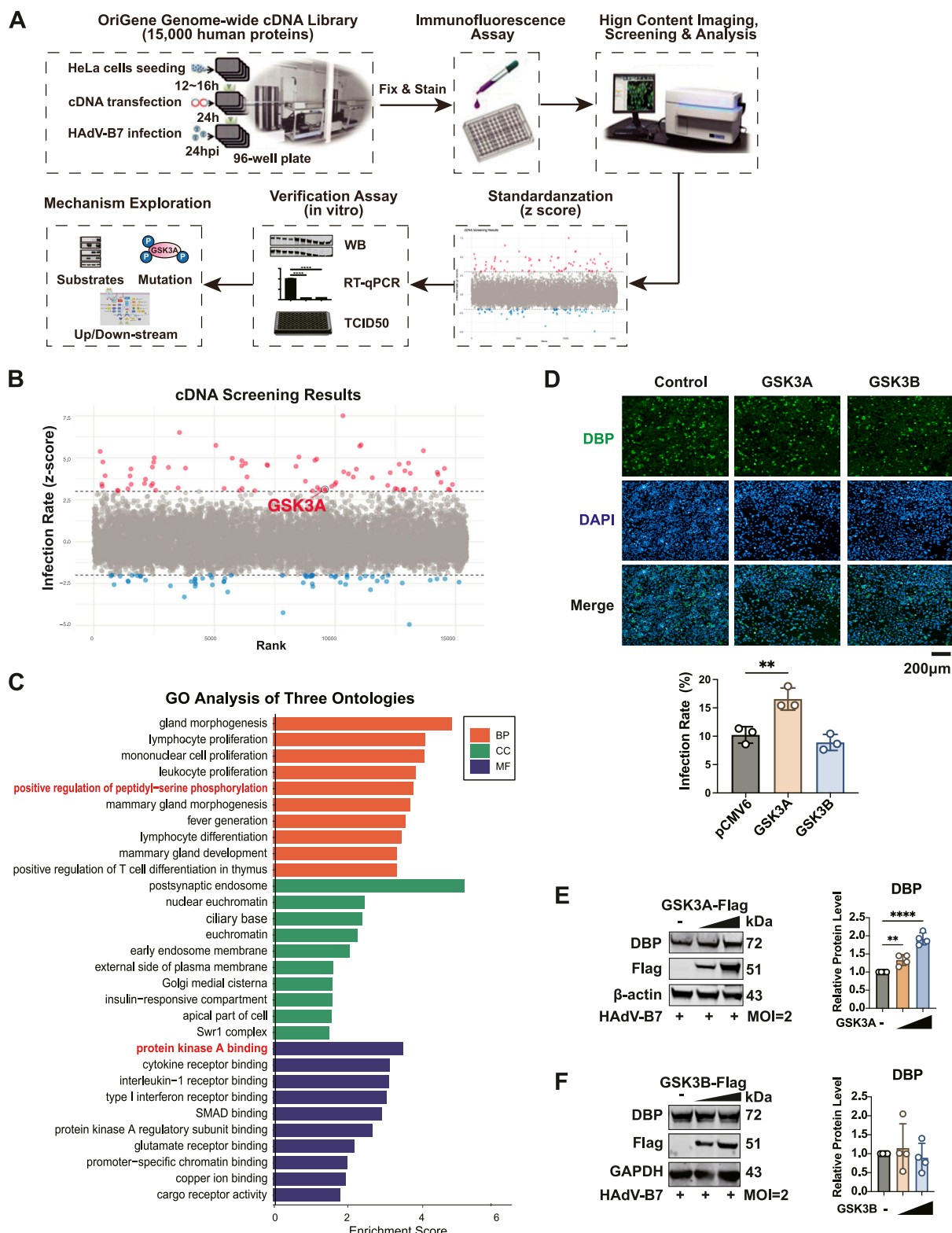

**Figure 1. Key host factors regulating HAdV-B7 were identified based on a genome-wide cDNA library screening method.**
**(A)** Schematic overview of the cDNA screening process using an OriGene genome-wide human cDNA library in HeLa cells. Cells were transfected with cDNA constructs, infected with HAdV-B7 (MOI = 10), and subjected to immunofluorescence-based high-content imaging for infection rate analysis. The infection rate was standardized using Z-score normalization, and hits were further validated through in vitro assays, including Western blot (WB), RT–qPCR, and TCID$_{50}$ titration. **(B)** Candidate proviral host factors identified with an absolute Z-score higher than 3 (marked in red), and antiviral host factors with an absolute Z-score lower than –2 (marked in blue). Nonsignificant candidates with an absolute Z-score between –2 and 3 were marked in gray. **(C)** GO function analysis of the identified host factors regulating HAdV-B7

in HeLa cells, this effect was absent in HeLa-GSK3A-KO cells (Fig 3I). The results indicated that the up-regulation of viral *hexon* transcripts is likely to be dependent on GSK3A, as GSK3B alone was insufficient to compensate for impaired viral replication caused by GSK3A depletion.

Collectively, these findings suggested that GSK3A uniquely functioned as a critical proviral host factor that enhanced viral replication and production.

## GSK3A depletion potently restricts the replication of the Species B epidemic strains

To determine the broad-spectrum role of GSK3A in the replication of HAdVs, we examined the replication of various HAdV strains spanning from Species B to E, which were isolated from cases of acute respiratory tract infection, in HeLa-GSK3A-KO cells. Cells were infected with various types of HAdV of strain at an MOI of 1 for 24 h, followed by Western blotting analysis of the viral DBP, RT–qPCR analysis of viral *DBP* transcripts, and measurement of viral titers in culture supernatants by the $TCID_{50}$ assay. The results indicated that GSK3A knockout markedly reduced DBP protein expression levels, with the most prominent effect in Species B strains (Fig 4A). A heatmap of normalized gray values from three independent WB replicates further illustrated the marked reduction in DBP protein levels across Species B~E (Fig 4B). Similarly, RT–qPCR analysis revealed a widespread decrease in viral *DBP* transcript levels, particularly affecting Species B~E upon GSK3A knockout (Fig 4C). In addition, viral titers of most HAdV strains were significantly reduced in the absence of GSK3A (Fig 4D). Notably, although Species B14 and C2 exhibited decreased viral titers in GSK3A-KO cells, these reductions were not statistically significant. Together, these findings suggested that GSK3A is a crucial pan-proviral host factor that facilitates HAdV replication across Species B~E.

## GSK3A's proviral effects are independent of HAdV binding and internalization

To identify the phase of the viral life cycle during which GSK3A exerts its effects, we examined the impact of GSK3A KO on viral binding and internalization of HAdV-B7 at an MOI of 10. Quantitative analysis of the viral gene revealed no significant differences in viral binding or internalization between HeLa-GSK3A-KO and WT cells (Fig 5A). Furthermore, a substantial decrease in viral *DBP* mRNA after GSK3A depletion was observed in both of GSK3A-KO cells at 8 and 12 h postinfection (hpi) using the RT–qPCR assay (Fig 5B). Given that a full replication cycle of HAdV-B7 typically completes within 12 h, GSK3A specifically promotes viral replication during the intermediate-to-late phase of the replication cycle (Pied & Wodrich, 2019).

## GSK3A kinase activity partially enhances HAdV-B7 replication

GSK3A, a dual-specificity kinase, is finely regulated by phosphorylation at key residues. Its activity is enhanced by selective phosphorylation of Tyr279 (Y279), whereas Lys148 (K148) contributes to activation under cellular stress (Buescher & Phiel, 2010). In contrast, phosphorylation at the N-terminal Ser21 (S21) inhibits GSK3A function by forming a primed pseudo-substrate and blocking further substrate binding at the catalytic groove (Doble & Woodgett, 2003). To investigate GSK3A's role in HAdV-B7 replication, S21A (nonphosphorylatable, constitutively active), Y279A (activation-deficient), and K148A (catalytically inactive) mutants were generated, and re-expressed in HeLa-GSK3A-KO1 cells (Fig 6A). The S21A mutant, representing a constitutively active form, significantly increased both viral DBP expression (Fig 6B) and viral titers (Fig 6C). In contrast, the Y279A and K148A mutants, both kinase-inactive forms, exhibited reduced DBP levels and viral titers compared with that induced by WT GSK3A (Fig 6B and C). However, viral titers in cells expressing the inactive mutants remained higher than those in the negative control group. These findings suggest that GSK3A promotes HAdV-B7 replication and viral production in a partially kinase-dependent manner.

## Depletion of GSK3A down-regulated viral protein phosphorylation levels

Recent studies found that GSK3B can stimulate the phosphorylation of the N protein of SARS-CoV-2, therefore enhancing the interaction between viral N and Nsp3 proteins (Yaron et al, 2022; Lin et al, 2023). Inspired by this, we performed a phosphoproteomic analysis to explore the possible viral peptides that can be down-regulated by GSK3A depletion (workflow seen in Fig 6D). Given that GSK3A specifically modifies serine and threonine residues, our data suggest a preference for serine phosphorylation over threonine in HAdV-B7 viral proteins (93% versus 7%, Fig 6E). By phosphopeptide enrichment, a total of 76 phosphorylation modification sites on almost 20 viral proteins were identified. The number of unique phosphorylation sites identified in HeLa-GSK3A-KO cells was reduced, compared with WT cells, with 15 sites identified exclusively in HeLa-GSK3A-KO cells versus 35 in WT cells (Fig 6E). Notably, 26 phosphorylation sites were shared between both groups (Fig 6E). Further quantification analysis identified 13 phosphorylation sites on 8 viral proteins significantly down-regulated by GSK3A deletion. To verify interactions between GSK3A and viral proteins, we carried out a co-IP analysis and identified three key viral proteins that interacted with GSK3A: L4-22K, E1A-I, and DBP. The phosphorylation landscape of these proteins is depicted in Fig 6F, and the corresponding secondary peak maps of mass spectrum analysis are shown in Fig 6G. Notably, phosphorylation at S105 on the peptide

replication included biological process (BP), molecular function (MF), and cellular component (CC). Enrichment scores are plotted to highlight the most significant results. **(D)** Immunofluorescence data from cDNA screening. The positive infection rate across 3 random fields was calculated by counting the number of GFP-positive cells relative to the number of DAPI-stained nuclei. **(E, F)** Verification of the regulatory roles of the candidate host factors on HAdV-B7 infection by Western blotting. HeLa cells were overexpressed with 0, 250, 500 ng Flag-tagged GSK3A or GSK3B (pCMV6-entry as control) for 24 h, then infected with HAdV-B7 (MOI = 2) for 36 h. Then, cells were lysed and immunoblotted for detection of viral DBP protein level (normalized by β-actin or GAPDH). Data information: in (D, E, F), data are presented as the mean ± SEM. *$P ≤ 0.05$ (*t* test).

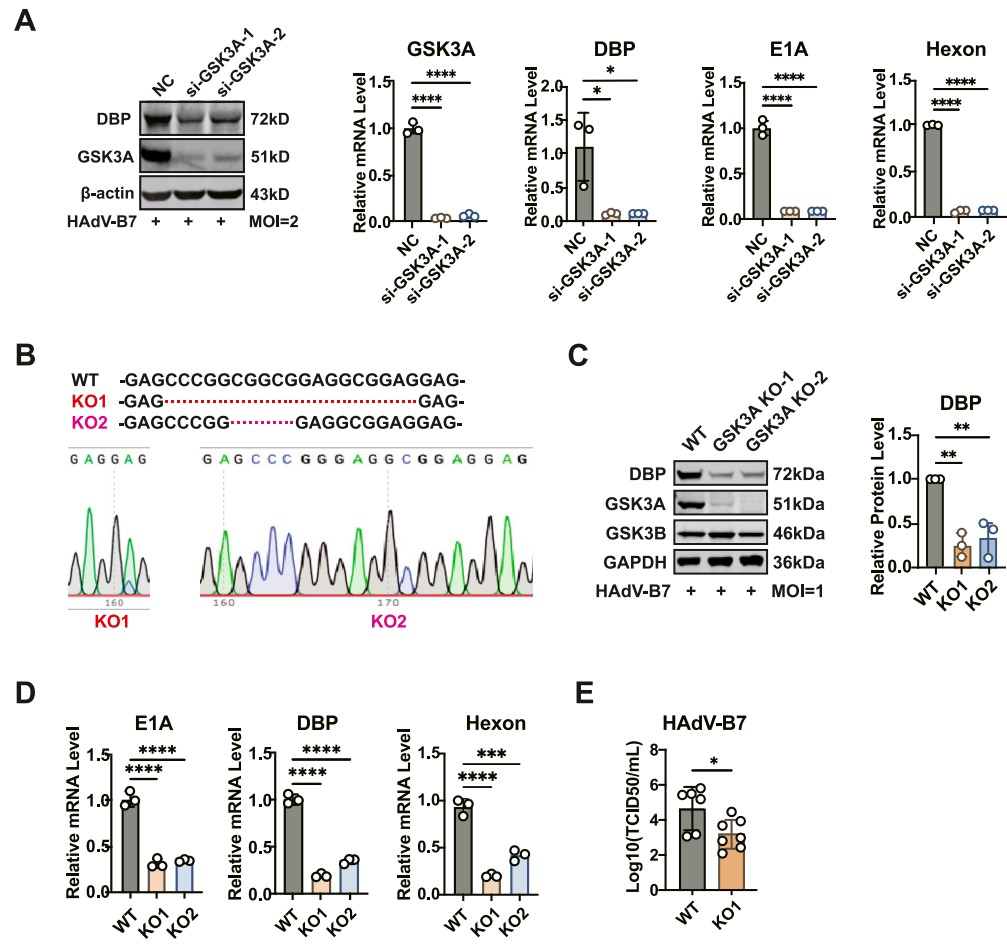

**Figure 2. GSK3A was required for HAdV replication.**
**(A)** HeLa cells were transfected with two different siRNAs targeting GSK3A for 48 h, followed by infection with HAdV-B7 (MOI = 2) for 36 h. Viral DBP protein levels were analyzed by immunoblotting, and viral *E1A*, *hexon*, and *DBP* transcript levels were quantified by qPCR. **(B)** Two HeLa-GSK3A-KO cell clones were generated using CRISPR-Cas9 and validated by sequencing. **(C)** WT and HeLa-GSK3A-KO cells were infected with HAdV-B7 (MOI = 1) for 36 h. GSK3A protein expression was confirmed by immunoblotting, and viral DBP protein levels were analyzed by immunoblotting. **(C, D)** WT and HeLa-GSK3A-KO cells were infected as in (C), and viral *E1A*, *hexon*, and *DBP* transcript levels were measured by RT–qPCR. **(C, E)** WT and HeLa-GSK3A-KO cells were infected as in (C), and the supernatant was collected to determine viral titers using the TCID$_{50}$ method. Data information: in (A, C, D, E), data are presented as the mean ± SEM. *$P \leq 0.05$ (*t* test).

"CYEEGFPPSDDEDGETEQSIHTAVNEGVK" of viral DBP, S125 on the peptide "HVPLQDIGHDSEEEREQAQLVAVGFSYPPVR" of viral E1A-I, and S78/S81 on the peptide "QLSSAAETSKSPDSSTATISAPGR" of viral L4-22K/33K was markedly down-regulated in HeLa-GSK3A-KO cells compared with WT cells (Fig 6H). Based on quantitative analysis, phosphorylation sites that were significantly reduced upon GSK3A deficiency are demonstrated in Fig 6H. The interactions between GSK3A and viral proteins E1A-I and L4-22K were validated by exogenous co-IP analysis (Fig 6I and J), whereas the interaction with DBP was confirmed via endogenous IP analysis (Fig 6K).

### GSK3A phosphorylates the viral L4-22K through its kinase domain

To investigate whether GSK3A kinase activity contributes to viral protein phosphorylation, we performed a co-IP assay between GSK3A and its mutants with L4-22K. Phosphorylation levels of L4-22K (L4-22K-p) were detected by immunoblotting with pan-serine/threonine antibody. A significant increase of the viral L4-22K phosphorylation level was found in the existence of GSK3A ectopic expression and S21A mutants compared with vectors (Fig 7A). In addition, the relative L4-22K-p level was low when inhibitory GSK3A mutations Y279A and K148A were present (Fig 7A). The results indicated that GSK3A's kinase activity, regulated by phosphorylation at residues K148 and Y279, plays a crucial role in modulating L4-22K phosphorylation. Simultaneously, to elucidate the impact of the GSK3A mutants on the binding affinity between the two proteins, we quantify this using a three-step calculation. First, the IP-enriched bands of each mutant were first normalized to their respective input levels to account for loading differences. Second, these values were then divided by the normalized L4-22K IP signal to control for variations in immunoprecipitation efficiency. Third, the binding affinities were standardized by the vector condition. Accordingly, the interaction between GSK3A and the viral L4-22K protein seemed to be increased by S21A mutation while weakened by K148A and Y279A mutations, consistent with the change in L4-22K-p levels (Fig 7A).

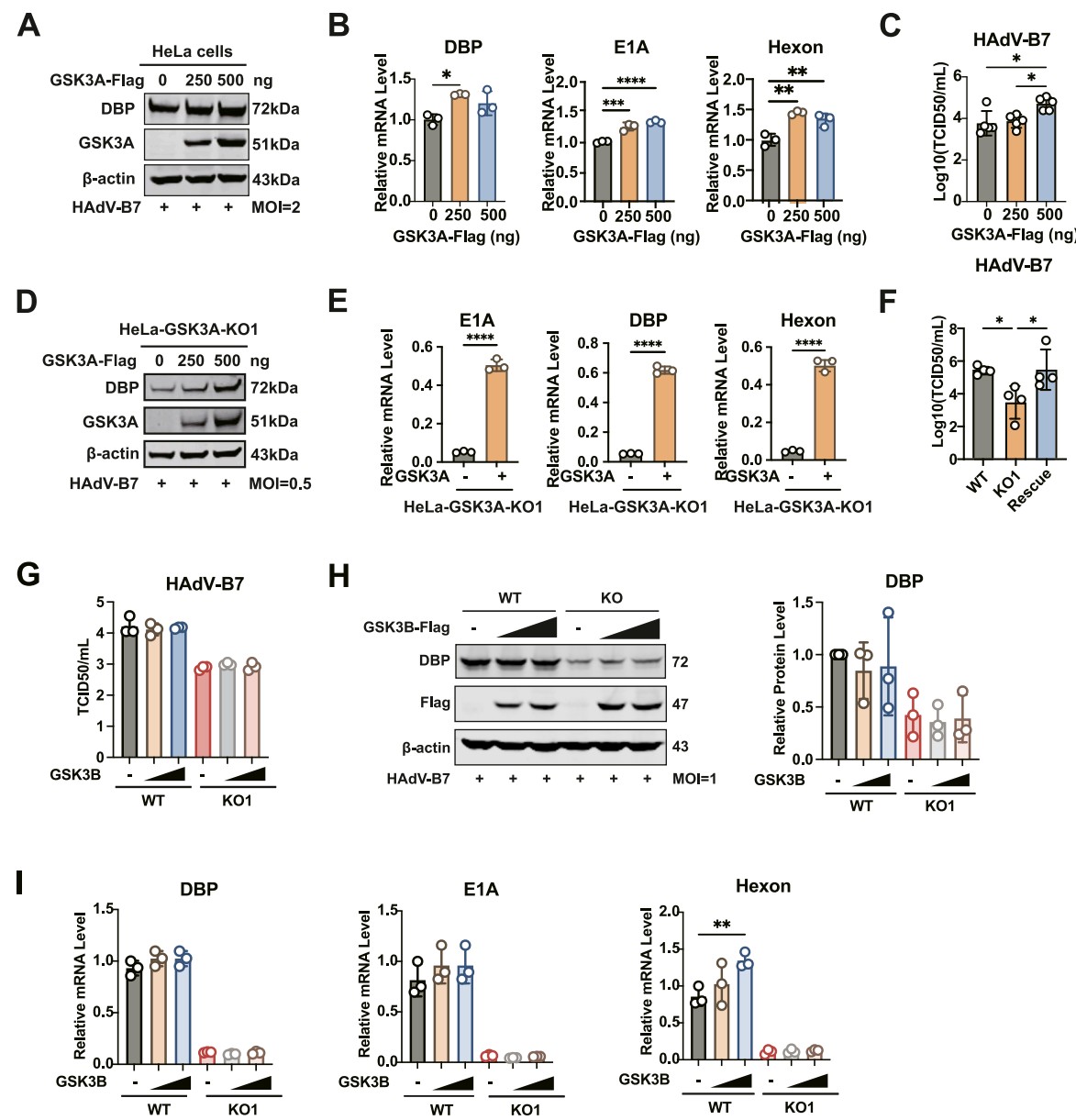

**Figure 3. GSK3A is essential for HAdV-B7 replication.**
**(A)** HeLa cells were transfected with 0, 250, or 500 ng of GSK3A-Flag plasmids for 24 h, followed by HAdV-B7 infection (MOI = 2) for 36 h. Viral DBP protein levels were analyzed by immunoblotting from three independent experiments. **(A, B)** HeLa cells were transfected and infected as described in (A), and viral *E1A*, *hexon*, and *DBP* transcript levels were quantified by RT–qPCR. **(A, C)** HeLa cells were transfected and infected as in (A), and the supernatant was collected to determine viral titers using the TCID$_{50}$ method (N = 5). **(D)** GSK3A was rescued by transfection of GSK3A-myc-flag plasmids in HeLa-GSK3A-KO1 cells for 24 h. Then, they are infected with HAdV-B7 (MOI = 0.5) for 36 h. The viral DBP protein levels were detected by immunoblotting (normalized by β-actin). **(D, E)** HeLa-GSK3A-KO1 cells were transfected and infected as in (D), and viral *E1A*, *hexon*, and *DBP* transcript levels were measured by RT–qPCR (normalized by β-actin). **(D, F)** HeLa-GSK3A-KO1 cells were transfected and infected as in (D), and the supernatant was collected to determine viral titers using the TCID$_{50}$ method (N = 4). **(G)** HeLa cells and HeLa-GSK3A-KO1 were transfected with 0, 250, or 500 ng of GSK3B-Flag plasmids for 24 h, followed by HAdV-B7 infection (MOI = 1) for 24 h. Viral titers of culture supernatant were measured by the TCID$_{50}$ method. **(G, H)** Same experiment treatment as in (G). Viral DBP protein levels (normalized by β-actin) were analyzed by immunoblotting from three independent experiments. **(G, I)** Same experiment treatment as in (G). Viral *DBP*, *E1A*, and *hexon* transcript levels were quantified by RT–qPCR analysis (normalized by β-actin). Data information: in (B, C, E, F, G, H, I), data are presented as the mean ± SEM. *P ≤ 0.05 (*t* test).

Consistent with our previous findings, deletion of the kinase domain (T2, spanning 114–406 aa) in GSK3A disrupted its interaction with L4-22K (Fig 7B), further supporting the regulatory role of GSK3A in L4-22K phosphorylation. To investigate the structural basis of this interaction, we predicted the structure of L4-22K using Google

ColabFold (v1.5.5) (Jumper et al, 2021). Subsequently, a protein–protein docking analysis was performed on the ClusPro 2.0 platform to visualize the binding pattern between GSK3A (PDB: 7SXF) and L4-22K (Jones et al, 2022; Amaral et al, 2023). In the interaction model, GSK3A was colored in pink with the predicted binding area (125–323

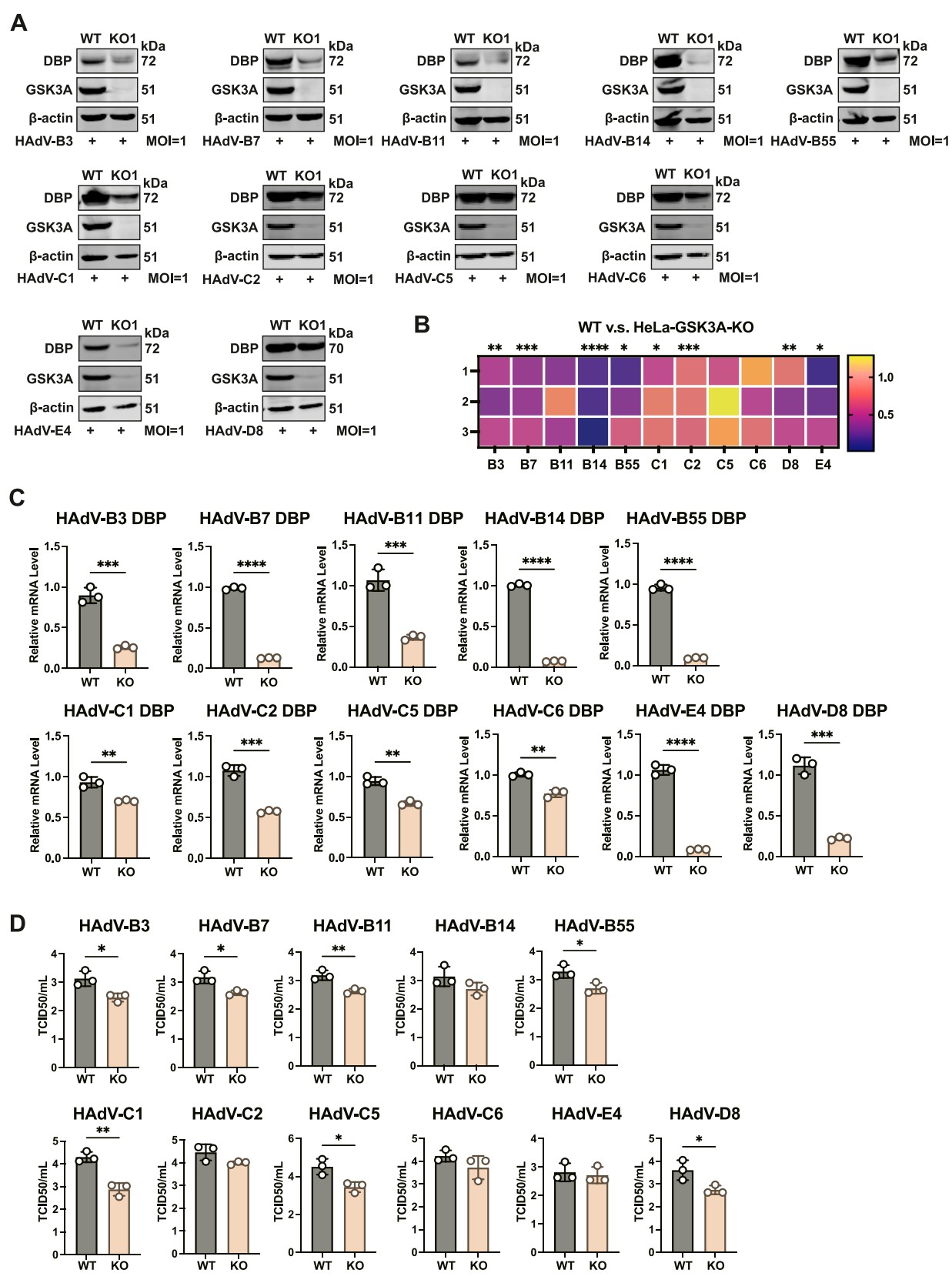

aa) located within the kinase domain highlighted in red (Fig 7C). Accordingly, the predicted binding interface of L4-22K to GSK3A was highlighted in yellow (92–96 aa, 98–104 aa, 121–126 aa, and 158–168 aa), with the phosphorylation sites S78 and S81 labeled for clarity (Fig 7C). To experimentally validate the binding region of L4-22K, we generated five truncation constructs: D1~4 (92–168 aa), D1 (92–96 aa), D2 (98–104 aa), D3 (121–126 aa), and D4 (158–168 aa). Co-IP analysis revealed that only deletions of 92–168 aa disrupted the interaction between L4-22K and GSK3A (Fig 7D), which is consistent with the protein–protein docking results. Notably, the phosphorylation sites (S78 and S81) lie outside the binding interface, suggesting that additional conformational changes may occur upon binding to GSK3A, potentially priming L4-22K for phosphorylation. Upon introducing single-point (S78A, S81A) and double-point (S78A/S81A) mutations, we performed a co-IP analysis with host protein GSK3A. The results indicated that the interaction between GSK3A and viral L4-22K protein persisted (Fig 7E), consistent with the predictions.

### Conservation of viral L4-22K S78/81 residues determines the GSK3A-mediated proviral phenotype during HAdV infection

To elucidate the mechanism behind the differential effects of GSK3A deficiency on various HAdV strains, we focused on the phosphorylation status of the L4-22K protein. Specifically, the suppression of L4-22K S78/81 phosphorylation of HAdV-B7 under GSK3A depletion suggested a regulatory role in viral replication. Considering this, we proposed that the conservation of specific L4-22K sequences might explain the broad-spectrum phenotypes of GSK3A deletion on various respiratory infectious HAdV strains. By analyzing the sequences of the L4-22K protein across various HAdV types, we observed that the S81 site is conserved only in HAdV-B3 (taking HAdV-B7 for reference, Fig 8A). Similarly, all B Species HAdVs exhibited high conservation of the S78 site, distinctly from Species C to E (Fig 8A). A phylogenetic analysis revealed the shortest genetic distance between HAdV-B3 and HAdV-B7, then went with other HAdV-B strains such as B11, B14, and B55, aligning with the observed inhibitory efficacy of GSK3A deficiency on the viral DBP expression level of these strains (Fig 8B). Combined with previous findings, we speculated that GSK3A-mediated phosphorylation of the L4-22K protein at the S78 and S81 residues might play a pivotal role in supporting HAdV replication.

## Discussion

Using a high-throughput cDNA library screening methodology containing 15,000 genes encoding host proteins, we identified GSK3A as a crucial proviral host factor during HAdV-B7 infection,

partially dependent on its kinase-regulatory function. In addition, we confirmed the unique proviral function of GSK3A while demonstrating no equivalent activity for its isoform GSK3B with overexpression verification experiments. The two isomers share high similarity in their kinase domains but differ in their N- and C-terminal regions (Duda et al, 2020). Unlike GSK3B, which shuttles between the cytoplasm and the nucleus, GSK3A is exclusively localized in the cytoplasm (Azoulay-Alfaguter et al, 2011). In many cases, they played overlapping negative regulatory roles in the WNT/β-catenin pathway, microtubule trafficking, glycogen synthesis, etc. (Cohen & Frame, 2001; Wu & Pan, 2010; Duda et al, 2020; Wang et al, 2022). Although often collectively referred to as GSK3 with presumed interchangeable functions, emerging evidence highlights key functional differences between them, particularly in antitumor therapies (Grassilli et al, 2014; Duda et al, 2020). It seems that GSK3B plays a more critical role in survival, as GSK3B KO embryos are lethal, whereas GSK3A KO mice remain viable (Hoeflich et al, 2000). Previous studies have linked GSK3A to neurodegenerative disorders and various cancers, including lung cancer (Greber & Flatt, 2019; Draffin et al, 2021; Cao et al, 2022). However, its role in viral infections, particularly HAdV, has not been explored. In this study, GSK3B showed no influence on HAdV-B7 replication, despite its reported proviral effects in hepatitis virus, influenza virus, and SARS-CoV-2 (Alfhili et al, 2020; Lin et al, 2023; Nishitsuji et al, 2025). Notably, GSK3A appeared to uniquely promote HAdV-B7 infections. This discrepancy may arise from structural differences between GSK3A and GSK3B, although the underlying mechanisms remain unclear and merit further investigation.

As a serine/threonine kinase, GSK3A phosphorylates various substrates, such as β-catenin, glycogen synthase 1 or 2, tau, playing critical roles in cellular signaling pathways (Wu & Pan, 2010; Greber & Flatt, 2019; Wang et al, 2022). Phosphorylation is a key posttranslational modification that regulates numerous physiological processes, often acting as a transient and cascade-like reaction. GSK3A is frequently involved in phosphorylation cascades, where it serves as an intermediate kinase. For instance, stimuli like nutrients or insulin activate the PI3K-AKT pathway, which in turn phosphorylates GSK3A at S21, rendering it inactive (Greber & Flatt, 2019). After ruling out any effects of GSK3A on viral entry, we found that GSK3A depletion restricted the transcript levels of a wide range of viral genes, including early genes *E1A*, *DBP* and late genes *hexon* at 8–12 hpi. Next, by generating several mutants, we demonstrated that the kinase activity of GSK3A, regulated by phosphorylation at S21, K148, and Y279, is essential for HAdV-B7 replication. Our results demonstrate that although WT GSK3A effectively promotes HAdV viral production, the kinase-active mutants enhanced viral replication and production to a significant extent compared with the baseline stimulation. The inactive mutants eliminated this promotion effect in viral replication but had less announced on viral production. This suggests that GSK3A's function in supporting viral

---

**Figure 4. Broad-spectrum inhibition of respiratory HAdV replication by GSK3A KO across Species B~E.**
HeLa-GSK3A-KO1 and WT cells were infected with different HAdV species (MOI = 1), including HAdV-B (types 3, 11, 14, 55), HAdV-C (types 1, 2, 5, 6), HAdV-D (Type 8), and HAdV-E (Type 4). Cells were lysed at 24 hpi, and viral DBP protein levels were analyzed by immunoblotting. t tests were used for statistical analysis, and the significance levels were labeled accordingly. **(A)** WB analysis of viral DBP protein levels across these HAdV strains. **(B)** Heatmap representing normalized gray values of viral DBP protein from 3 replicates, adjusted by β-actin levels. **(C)** RT–qPCR quantification of viral *DBP* transcript levels across these HAdV strains. **(D)** Viral titers across these HAdV strains measured by $TCID_{50}$ assays. Data information: in (B, C, D), data are presented as the mean ± SEM. *$P \leq 0.05$ (t test).

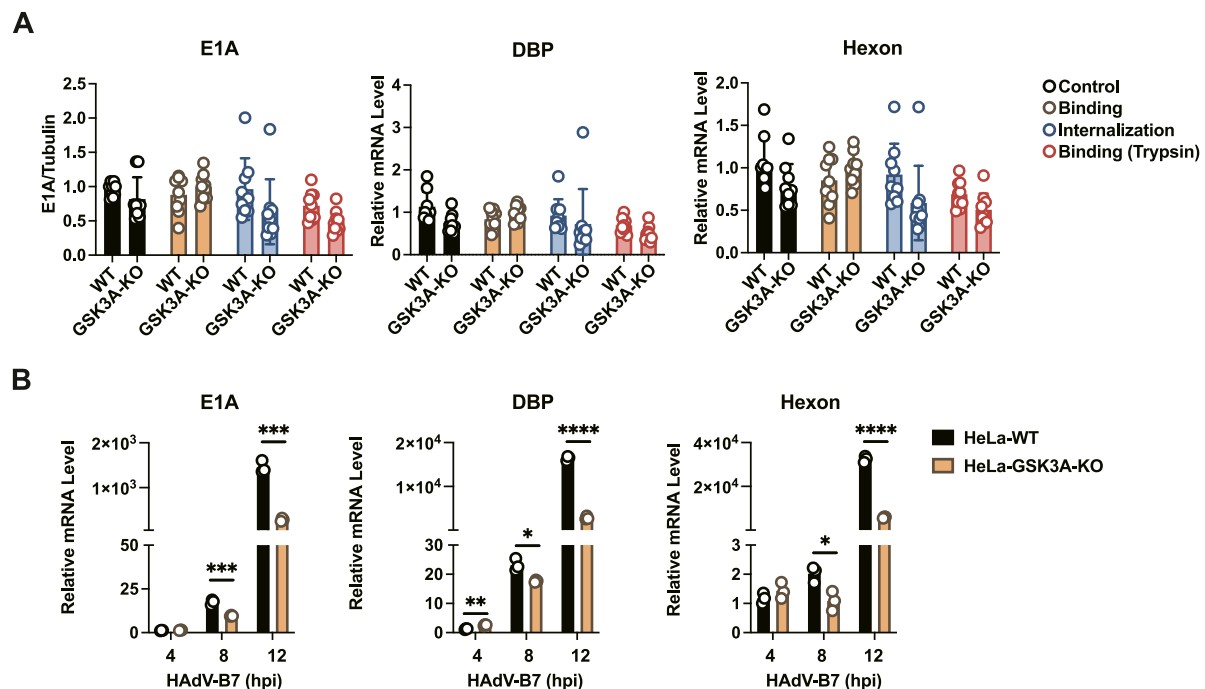

**Figure 5. GSK3A acts independently of viral entry.**
**(A)** HeLa cells were infected with HAdV-B7 (MOI = 10) under three conditions: binding group (incubated at 4°C for 1 h), internalization group (incubated at 4°C for 1 h followed by 37°C for 30 min), and control group (maintained at 37°C). The sample size was N = 9 for each group. Viral DNA were collected and the levels of *E1A*, *DBP*, and *hexon* were quantified by fluorescent quantitative PCR analysis. **(B)** WT and HeLa-GSK3A-KO cells were infected with HAdV-B7 (MOI = 1), and intracellular viral *E1A*, *DBP*, and *hexon* transcript levels were measured by RT–qPCR. The sample size was N = 3 for each group. Data information: in (A, B), data are presented as the mean ± SEM. *$P \leq 0.05$ (t test).

replication is partially dependent on its kinase activity, but the kinase-independent function cannot be excluded. However, an unexpected increase in virus production was observed in the inactive mutant group, compared with negative controls, suggesting that nonenzymatic roles of GSK3A (potential structural or scaffolding roles such as stabilizing viral proteins or host factors based on physical contacts) might exist to facilitate virus assembly or release, thereby contributing to elevated viral yields. The precise mechanisms underlying these kinase-independent roles warrant further investigation in future studies.

Because GSK3A was found to phosphorylate viral proteins (N protein of SARS-CoV and SARS-CoV-2), we hypothesized it would phosphorylate adenoviral proteins directly and affect their functions (Wu et al, 2009; Yaron et al, 2022; Lin et al, 2023). Previous studies have suggested the possibility for phosphorylation modification to occur on viral proteins of HAdV, such as E1A, DBP, and DNA polymerase (Russell et al, 1989; Dumont & Branton, 1992; Ramachandra et al, 1993). It has been shown that phosphorylation of adenovirus DNA polymerase is crucial for its bioactivity (Ramachandra et al, 1993). After the phosphoproteomic analysis and co-IP experiments, we identified viral L4-22K as a key substrate of GSK3A that contributes to its proviral activities. Specifically, phosphorylation at S78/81 residues of L4-22K of HAdV-B7 was significantly reduced after GSK3A depletion. Moreover, the fluctuations in the phosphorylation levels of L4-22K correspond to changes in GSK3A kinase activities regulated by residues K148 and Y279. However, the activation of S21A mutation of GSK3A did not cause a further increase in L4-22K-p levels compared with WT

GSK3A, possibly because of insufficient ATP availability or substrate saturation. Our findings indicated that GSK3A functions by stimulating L4-22K phosphorylation, which is partially dependent on its kinase activity. Previous studies predicted potential phosphorylation of HAdV-C5 L4-22K by various kinases, including PKA, creatine kinase (CK), CDK, AKT, GSK3 (Biasiotto & Akusjärvi, 2015). Here, we provided evidence for GSK3A stimulating L4-22K phosphorylation based on co-IP analysis, but lack of in vitro biochemical verifications. Therefore, we cannot deny the possibility of endogenous factors participating in this process. Further studies in vitro are required to confirm GSK3A's regulation on adenoviral proteins including L4-22K.

To understand this further, we mapped the exact binding interface of GKS3A and viral L4-22K proteins with truncation experiments based on protein structure prediction and protein–protein docking. However, the structural modeling using AlphaFold2 revealed that regulated S78/81 sites do not overlap with the interaction region. Considering that GSK3A often recognizes prephosphorylated substrates, a conformational change may occur after its interaction with L4-22K, to allow S78/81 phosphorylation to occur (Beurel et al, 2015; Greber & Flatt, 2019). Or other co-regulatory kinases primarily phosphorylated S78 residues and induced a conformational change; therefore, the L4-22K S78-p was primed to be phosphorylated by GSK3A, similar to GSK3B (Sutherland, 2011). However, the functional implications of these phosphorylation events in HAdV-B7 replication remain unelucidated. Previous studies indicated a complex function of the L4-22K protein in viral major late promoter transcription (low L4-22K level

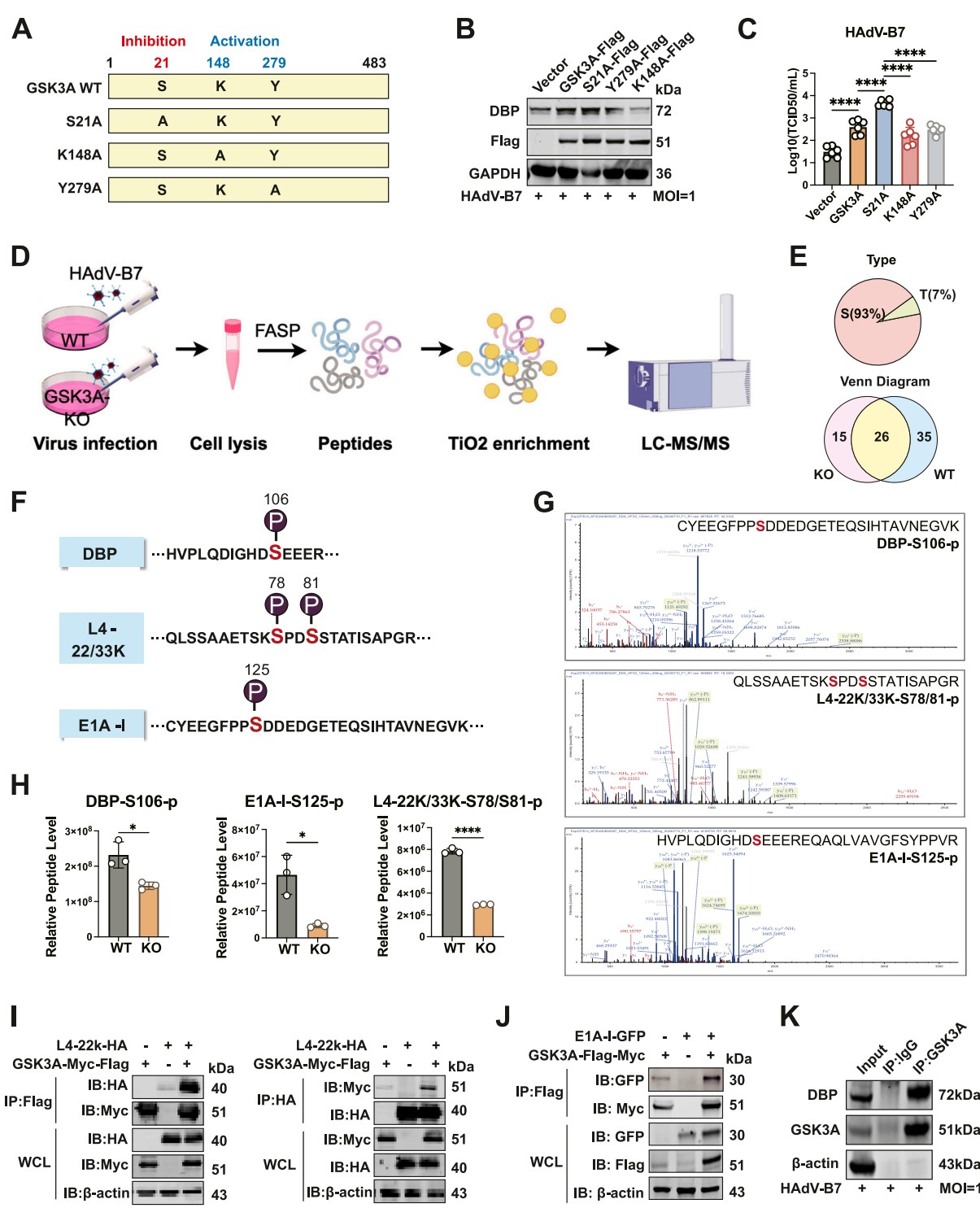

**Figure 6. Identification of GSK3A-regulated viral phosphorylation sites by phosphoproteomic analysis.**

**(A)** Schematic representation of three phosphorylation sites in GSK3A. S21 is an inhibitory site, where phosphorylation suppresses kinase activity, whereas K148 and Y279 are activation sites, whose phosphorylation indicates an active kinase state. **(B)** HeLa-GSK3A-KO cells were transfected with WT or point-mutant GSK3A (Flag-Myc tagged) plasmids for 24 h, followed by HAdV-B7 infection (MOI = 1) for 36 h. Cellular viral DBP protein levels and exogenous GSK3A expression (anti-Flag) were assessed by immunoblotting. **(C)** HeLa-GSK3A-KO cells were transfected with WT or point-mutant GSK3A (Flag-Myc tagged) plasmids for 24 h, followed by HAdV-B7 infection (MOI = 1) for 36 h. Viral titers in the supernatant were measured using the $TCID_{50}$ assay (N = 7/group). **(D)** Workflow of phosphoproteomic analysis was drawn in FigDraw. WT and HeLa-GSK3A-KO cells were infected with HAdV-B7 (MOI = 2) for 24 h. Cell lysates were processed and digested into peptides, followed by $TiO_2$ enrichment, LC-MS/MS analysis, data processing, and viral protein sequence alignment to identify differentially expressed phosphorylated amino acid sites. The sample size was N = 3 for each group. **(E)** Pie chart of identified phosphorylated amino acid types regulated by GSK3A depletion. **(F)** Landscape of candidate phosphorylation sites on viral proteins of HAdV-B7, including DBP, L4-22K/33K, and E1A-I, regulated by GSK3A. Red indicated down-regulation of phosphorylation level in HeLa-GSK3A-KO cells ($P < 0.05$).

activated major late promoter transcription, whereas high level inactivated), mRNA splicing (e.g., L4-33K), viral genome packaging (e.g., 52K, pIVa2), viral RNA exporting (from the nucleus to cytoplasm), posttranscriptional steps (e.g., regulating hexon translation or protein stability), and antagonizing E2A (Backström et al, 2010; Guimet & Hearing, 2013; Lan et al, 2017; White et al, 2023). We failed to delete L4-22K in infectious HAdV-B7 variants or rescue viable virus with S78A or S81A mutations based on the reverse genetics approach, highlighting their essential role in virus replication and growth. Given that L4-22K played such an important role in multiple steps of virus replication, we proposed that GSK3A might affect its function, kinetics, or protein stability by stimulating S78 and S81 phosphorylation after a specific conformational change. However, further studies were required to address the function of L4-22K-p. For instance, phosphorylation of L4-22K may affect the protein activity, protein synthesis or degradation, the cellular location, the interaction of it with other host factors or viral proteins, etc. Particularly, the viral L4-22K proteins reside primarily in the nucleus (Soriano et al, 2019). Given GSK3A's cytoplasmic localization in published data, its stimulation of viral L4-22K protein phosphorylation might present a complex scenario (Azoulay-Alfaguter et al, 2011). We assumed that GSK3A phosphorylates newly synthesized viral L4-22K proteins in the cytoplasm before they were transported into the nucleus for progeny virion assembly. However, further research is required to elucidate the spatial and temporal dynamics of GSK3A-mediated phosphorylation of viral proteins like L4-22K and its impact on viral replication patterns.

On the other hand, the possibility of GSK3A phosphorylating host transcription factors and regulating viral gene transcriptions still exists. For instance, GSK3B can phosphorylate transcription factor upstream stimulatory factor 2, altering its conformational structure and affecting its DNA binding activities (Horbach et al, 2014). In our studies, GSK3A depletion restricted virus replication and production, as well as viral gene transcriptions (including early and late viral genes). But the underlying mechanisms were not clear.

In addition to elucidating the GSK3A's proviral activity on HAdV-B7 replication, we explored its impact on other HAdV types that infect the respiratory system, including HAdV Species B~E. By assessing viral titers, viral DBP protein, and transcript levels, we found that respiratory HAdVs were broadly restricted by GSK3A depletion, especially for Species B. Notably, HAdV-B, especially HAdV-B3 and HAdV-B7, has become the dominant species contributing to acute respiratory tract infection patients in China from 2009 to 2020 (Liu et al, 2023). The expression of viral DBP protein and mRNA levels of Species B HAdV in HeLa-GSK3A-KO cells remained the lowest compared with Species C and D. This suggests that GSK3A plays an essential role in multiple adenoviral types, making it an attractive target for antiviral drug development. Consequently, sequence alignment of the L4-22K protein across species reveals that these HAdV strains also exhibited the highest similarity among

Species B HAdV, with a conserved S78 residue. This suggested that the proviral efficacy of GSK3A might be linked to the phosphorylation on S78 of L4-22K. Accordingly, GSK3B preferentially phosphorylates substrates containing the "S/TXXXS/T" motif. As a result, we assumed that the "S/TXXXS/T" and "S/TXXS/T" motif of viral L4-22K protein across respiratory HAdV strains may explain the broad-spectrum inhibitory phenotype of GSK3A depletion on virus replication and production in Species B (Ter Haar et al, 2001).

In summary, GSK3A is a proviral host factor for HAdV-B7 infection, with its phosphorylation of viral proteins. GSK3A interacted with viral L4-22K through its kinase domain and stimulated viral L4-22K phosphorylation at S78/81. This process partially relies on GSK3A's kinase activity, regulated by its key phosphorylation sites at K148 and Y279. These findings provide a foundation for further exploration of the phosphorylation mechanism and the development of specific inhibitors, paving the way for potential therapeutic strategies against HAdV-B7 infections.

# Materials and Methods

## Cell lines and viruses

Cell lines, including HeLa, A549, and 293T cells, were all purchased from the ATCC, and maintained in DMEM supplemented with 10% FBS. HeLa cells with a GSK3A knockout (HeLa-GSK3A-KO) were produced via CRISPR-Cas9 genome editing. The small-guide RNA (sgRNA) sequences targeting GSK3A employed for this purpose were "GACTAGCTCGTTCGCGGAGCCCGG" and "AGCTCGTTCGCGGAGCCC GGCGG." All cell lines were incubated at 37°C in a humidified incubator with 5% $CO_2$. All cell lines were verified through PCR analysis to ensure they were *Mycoplasma*-free.

Various types of HAdV strains, including HAdV-B7, HAdV-B3, HAdV-B11, HAdV-B55, HAdV-C1, HAdV-C2, HAdV-C5, HAdV-D8, and HAdV-E4, were isolated from respiratory specimens of children with acute respiratory tract infection at Beijing Children's Hospital, Capital Medical University. Subsequently, these strains were purified three times using the plaque assay in A549 cells. HAdV viruses were propagated in A549 cells with DMEM containing 2% FBS. Viral titer was determined by the 50% tissue culture infection dose assay ($TCID_{50}$), as described below. The study protocol was reviewed and approved by the Ethical Review Committee of Beijing Children's Hospital, Capital Medical University (2017–k-15).

## High-throughput screening

HeLa cells were seeded into 96-well plates at a density of $1.0 \times 10^4$ cells per well and incubated for 16–18 h. Subsequently, cells were

---

(G) Representative MS/MS spectrum of the identified phosphopeptides in the phosphoproteomic analysis. The annotated peaks correspond to fragment ions used to localize the phosphorylation sites. The corresponding site-specific phosphorylation at the serine residues was marked in red at the top right of each peak map. The key b- and y-ions were labeled to support confident peptide sequence identification and phosphorylation assignment. (G, H) Quantitative analysis of the selected phosphopeptides by parallel reaction monitoring, corresponding to the peak maps in panel (G). (I) Co-IP analysis of GSK3A-myc-flag and L4-22K-HA in 293T cells. The left was enriched with Flag-tagged beads, whereas the right was enriched with HA-tagged beads. (J) Co-IP analysis of GSK3A-myc-flag and E1A-I-GFP in 293T cells. (K) IP analysis of GSK3A and viral DBP protein in HeLa cells infected with HAdV-B7 (MOI = 1) for 24 h. The cell lysate was incubated with anti-GSK3A antibodies using protein A/G beads. Data information: in (C, H), data are presented as the mean ± SEM. *$P \leq 0.05$ (*t* test).

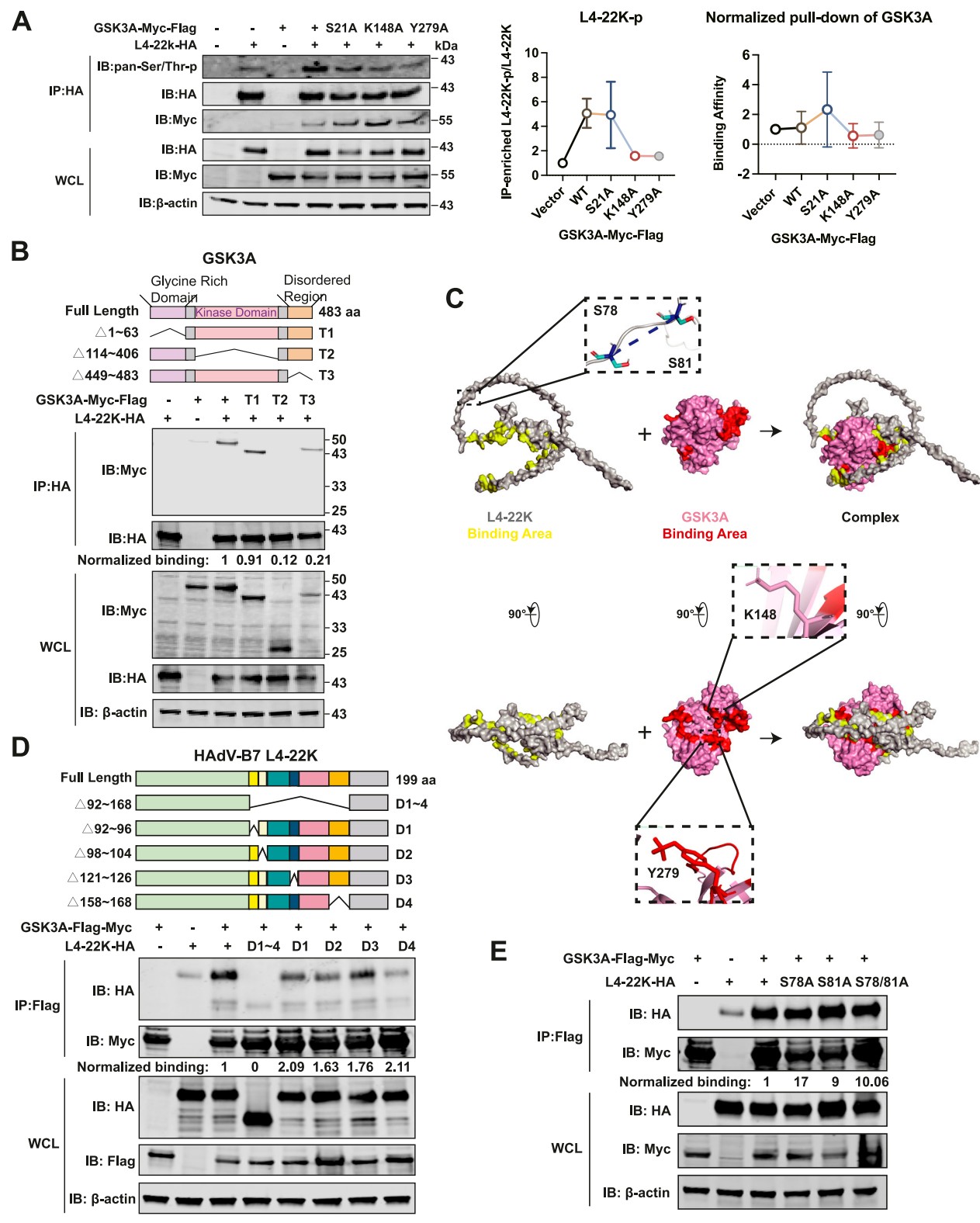

**Figure 7. GSK3A binds viral L4-22K protein through its kinase domain and phosphorylates L4-22K via kinase activity.**
(A) Co-IP kinase assay. 293T cells were co-transfected with WT or mutant GSK3A-flag-myc plasmids and L4-22K-HA plasmids for 24 h. L4-22K-HA protein was enriched in cell lysates using HA-tagged beads. The phosphorylation level of L4-22K (L4-22K-p) regulated by GSK3A was measured by immunoblotting with the anti-pan-Ser/Thr-phosphorylation antibodies. The relative L4-22K-p level was calculated by the gray value of the 40 kD phosphorylation band divided by that of the anti-HA band in IP samples (the total protein level of HA-tagged L4-22K). Statistical analysis was performed on data from three independent experiments. The band intensities of GSK3A mutants and L4-22K protein were measured and calculated by normalizing WT and GSK3A mutant IP signals to their input (also whole-cell lysates, WCL) levels, then

transfected with a human cDNA library comprising ~15,000 distinct full-length cDNA clones (TrueClone Collection; OriGene) at a concentration of 100 ng per well using FuGENE HD Transfection Reagent (Promega) for 24 h. An empty vector was used as the negative control.

The cells were then infected with HAdV-B7 at an MOI of 10. After 2-h incubation, the cells were washed once with Dulbecco's phosphate-buffered saline (DPBS), followed by the addition of DMEM containing 2% FBS. The cultures were then incubated for 48 h at 37°C in a 5% $CO_2$ incubator. Subsequently, cells were fixed with an equal volume of 8% PFA (paraformaldehyde) solution for 20 min. Then, a QuickBlock Blocking Buffer containing Triton X-100 (P0260; Beyotime) was added to perforate the membrane and block nonspecific bindings for 30 min. Anti-DBP (HAdV DNA binding protein; DBP) rabbit polyclonal antibody (1:5,000, from Beijing Children's Hospital) and anti-rabbit IgG with Alexa Fluor 488 Conjugate (1:1,000, 4412; CST) were subsequently employed for 1.5 h at room temperature to quantify virus replication using the immunofluorescence assay. Imaging was performed using an Operetta high-content imaging system (PerkinElmer), with subsequent analysis conducted using Harmony software to determine HAdV-B7 infection rates.

Infection rates from each sample were transformed into Z-scores, which reflect the number of SD by which the experimental infection rates deviated from the mean plate value. Wells exhibiting significant cytotoxicity, defined as a greater than threefold decrease in signal intensity, were excluded from further analysis. The threshold for significant infection rates was established at a Z-score higher than 3 and lower than –2.

### Plasmids

The following plasmids for cDNA screening were used and stored at the Christophe Merieux Laboratory: pCMV6-entry, psPAX2, pVSV-G, and pLenti-CRISPRv2. Additional plasmids, including GSK3A-myc-flag and GSK3B-myc-flag, were purchased from OriGene Technologies. The viral protein plasmids such as pcDNA3.1-L4-22K-HA, pcDNA3.1-DBP, and pEGFP-c1-E1A-I were constructed by Beijing Children's Hospital. Mutants of GSK3A or truncated GSK3A or viral L4-22K were introduced by a PCR-mediated site-directed mutagenesis assay using KOD One PCR Master Mix (KMM-101; TOYOBO) and Dpn I (R0176S; New England Biolabs), which was previously described, and verified by Sanger sequencing (Carey et al, 2013).

### Western blot analysis (WB)

For WB analyses, cells were lysed in RIPA lysis buffer containing protein inhibitor cocktail (04693132001; Roche) on ice for 30 min. The lysate was then subjected to centrifugation at 12,000*g* for 10 min at 4°C to remove cellular debris. The concentration of total protein in the supernatant was quantified with Pierce BCA Protein Assay Kit (23227; Thermo Fisher Scientific). Thirty µg protein was mixed with 6×SDS–PAGE loading buffer and electrophoresed within a 10% SDS–PAGE gel, initially at 80V for 30 min and then at 120V for 1.5 h. Subsequently, proteins were transferred to a nitrocellulose (NC) membrane (Millipore) at a constant current of 250 mA for 2 h, with the transfer unit bathed in ice. After blocking with 5% skim milk or 5% BSA, the membrane was incubated with primary antibody overnight at 4°C, including anti-Flag (1:4,000, F3165; Sigma-Aldrich), anti-HA (1:4,000, H6908; Sigma-Aldrich), anti-Myc (1:4,000, A5598; Sigma-Aldrich), GSK3A (1:1,000, R1312-1; HUABIO), anti-DBP rabbit polyclonal antibody (1:5,000; prepared by Beijing Children's Hospital), and anti-pan-Ser/Thr phospho-antibody (1:500, SAB570025; Sigma-Aldrich). Three widely used reference proteins (β-actin [1:4,000, 66009-1-Ig; Proteintech], glyceraldehyde 3-phosphate dehydrogenase [GAPDH, 1:4,000, 10494-1-AP; Proteintech], and α-tubulin [1:4,000, 66031-1-Ig; Proteintech]) were employed as an internal control. Immunoblots were then visualized using Odyssey CLx Imaging System (LI-COR Biosciences) and analyzed in ImageStudio Software (v5.2.5; LI-COR 2014). Protein molecular weights were determined by comparison with pre-stained molecular weight markers (YEASEN, 20350ES).

### RT–qPCR analysis

Total RNA was purified using TRIzol Reagent (Invitrogen), followed by reverse transcription into cDNA using First-strand cDNA Synthesis Mix (F0202; LABLEAD) according to the manufacturer's instructions. The relative gene expression level was analyzed using SYBR Green dye (RR067A; TAKARA) and specific primer pairs on the real-time fluorescence quantitative PCR instrument (Bio-Rad). The expression levels of the host or viral mRNA were standardized relative to the housekeeping gene mRNA, GAPDH. The following primers were used for quantitative PCR analyses, including HAdV-B7 E1A forward: 5′-ATGGCTGTAAGTCTTGTGAA-3′, HAdV-B7 E1A reverse: 5′-AGGAGGTGAGGTAGTTGAA-3′, HAdV-B7 DBP forward: 5′-ATGGAAGTGATGGCTGTGCTAATGG-3′, HAdV-B7 DBP reverse: 5′-CTTGTACTGCTCGTGCTGCTCTG-3′, HAdV-B7 Hexon forward: 5′-CAGGTGGTTGATGAGGTTA-3′, HAdV-B7 Hexon reverse: 5′-GGCAGTAGTTCCGATGAG-3′, GSK3A forward: 5′-CAATATTGCAGTGGTCCAGC-3′, GSK3A reverse: 5′-GGGAACTAGTCGCCATCAAG-3′, β-actin forward: 5′-CATGTACGTTGCTATCCAGGC-3′, β-actin reverse: 5′-CTCCTTAATGTCACGCACGAT-3′.

### siRNA knockdown experiments

HeLa cells at a density of $8 \times 10^4$ were seeded in 24-well plates. After overnight incubation, transfection was carried out with 100

dividing by the normalized L4-22K protein IP signal, and standardized by the vector condition. **(B)** 293T cells were co-transfected with GSK3A truncations and L4-22K for 24 h. Cells were lysed and enriched using HA-tagged beads for co-IP analysis. Band intensities were analyzed as described in panel A, but standardized by the GSK3A-WT condition. **(C)** Prediction structure of viral L4-22K and protein–protein docking model of the GSK3A-L4-22K complex. The interaction interface was defined by atoms within 5 Å between GSK3A and L4-22K. The L4-22K protein was shown in gray, with the predicted binding region marked yellow. The GSK3A was shown in pink, with the predicted binding area highlighted in red. **(D)** Truncated L4-22 K HA–tagged plasmids were co-transfected with GSK3A double-tagged plasmid and enriched with HA-tagged beads for co-IP analysis. Band intensities of L4-22K truncations and GSK3A protein were quantified by normalizing WT and L4-22K mutant IP signals to their input levels and then dividing by normalized GSK3A-WT signals. **(E)** Co-IP analysis between S78A and S81A mutations of viral L4-22K protein and GSK3A protein. Band intensity quantification was performed similar to panel D for L4-22K mutants. Data information: in (A), data are presented as the mean ± SEM. *$P \leq 0.05$ (*t* test).

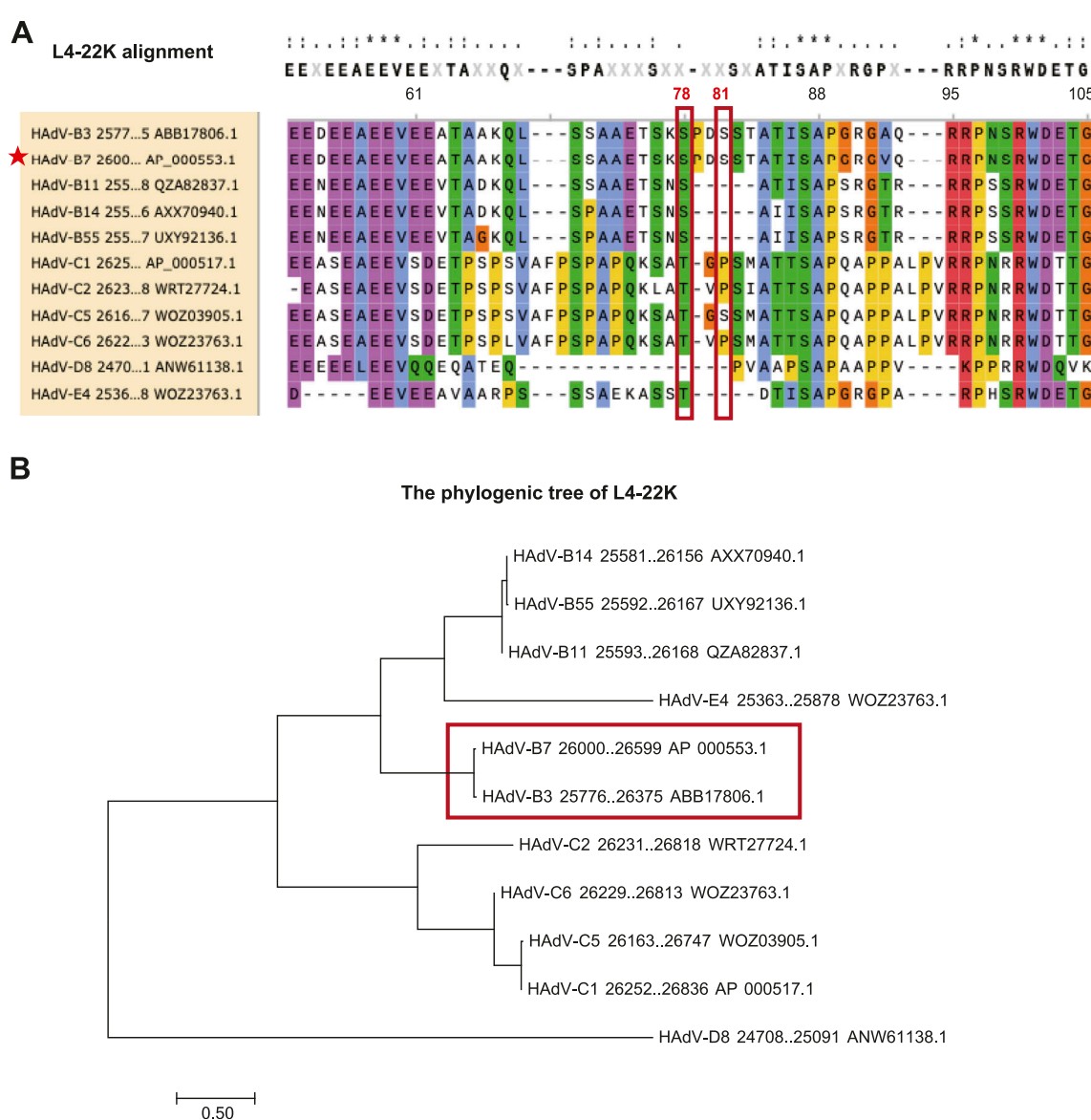

**Figure 8. Sequence analysis of L4-22K across HAdV Species B~E.**
**(A)** Sequence alignment of viral L4-22K proteins performed in SnapGene using the MAFFT method. That of HAdV-B7 was used as a reference sequence. L4-22K S78 and S81 of HAdV-B7 were marked in red boxes. **(B)** Phylogenetic tree was generated by MEGA7 across various respiratory HAdV strains including L4-22K sequences of HAdV-B3, HAdV-B7, HAdV-B11, HAdV-B14, HAdV-B55; HAdV-C1, HAdV-C2, HAdV-C5, HAdV-C6; HAdV-D8, HAdV-E4, etc. Sequences were retrieved from GenBank (before 28 October 2024).

pmol of siRNA (GenePharma) using 2 μl of Lipofectamine RNAiMAX Reagent (Thermo Fisher Scientific) for 6 h. Subsequently, the cells were rinsed with DPBS and further cultured in fresh DMEM (11995065; GIBCO) enriched with 10% FBS (35081; Corning). 48 h after transfection, the cells were inoculated with HAdV variants at an MOI of 2 for 2 h. After this period, the cells were collected, lysed, and processed for viral DNA extraction as outlined previously. The siRNA sequences targeting GSK3A were used as follows: the first siRNA, forward: 5′-CCCAAUGUCUCCUACAUCUTT-3′, reverse: 5′-AGAUGUAGGAGACAUUGGGTT-3′; the second siRNA, forward: 5′-CCUGGACAAAGGUGUUCAATT-3′, reverse: 5′-UUGAACACCUUUGUCCAGGTT-3′.

### Fifty percent tissue culture infectious dose (TCID50) assay

The viral titer is measured by the $TCID_{50}$ method, employing a cell-based immunofluorescence assay. The viral supernatant was centrifuged and collected after a half-logarithmic dilution method. A549 cells, previously seeded and incubated overnight in 96-well plates, were infected with various dilutions, with four replicates for each dilution. Cells were immobilized and subsequently examined for the presence of viral DBP expression using a primary antibody. Detection was achieved with an Alexa Fluor 488-conjugated secondary antibody (4412; CST) under a fluorescence microscope. More than three positive cells/pores were considered positive, and the

virus titer was calculated with the Reed and Muench method (Reed & Muench, 1938).

### Virus binding and internalization assay

HeLa cells, including both WT and GSK3A knockout cells (HeLa-GLaSK3A-KO), were seeded into 12-well plates at a density of $2 \times 10^5$ cells per well and incubated overnight. For the viral binding assay, these cells were inoculated to HAdV-B7 at an MOI of 10 in a cold medium without FBS on ice for 60 min. After this, the cells were washed three times with ice-cold DPBS with or without protease treatment and harvested, and the DNA from the attached viruses was extracted and quantified via qPCR. For the viral internalization assay, cells were first incubated with HAdV-B7 at an MOI of 10 on ice for 60 min, then washed thrice with cold DPBS, and subsequently treated with a pre-warmed medium containing FBS at 37°C for 30 min. After this, cells were cleaned with DPBS three times and harvested with a scraper. The internalized viruses, present in both endosomal and cytoplasmic compartments, were quantified using qPCR as previously described.

### Phosphoproteomics

The WT and HeLa-GSK3A-KO cells were infected with HAdV-B7 (MOI = 1) for 24 h, and then, the cells were collected and lysed in an 8 M urea buffer supplemented with protease/phosphatase inhibitor cocktails. After protein extraction, reduction (in 10 mM DTT), alkylation (in 20 mM IAA), and digestion with trypsin were performed using a filter-aided proteome preparation method. Then, the peptides were desalted and enriched using $TiO_2$ beads for 30 min. After two sequential elution steps, the phosphopeptides were separated in an EASY-nLC 1,200 nano-flow liquid chromatography system. Peptides were loaded on a trap column and separated on an analytical column at a flow rate of 600 nl/min. Analysis was performed on a Q Exactive HF-X mass spectrometer in positive mode. Full MS2 scans (300–1,400 m/z) were recorded at 120,000 resolutions, followed by MS2 scans for the top 20 ions at 15,000 resolutions using HCD (NCE 27%). Data were processed, aligned with the HAdV-B7 sequence from GenBank (database derived on 3 October 2024), and quantified with the iProteome one-stop cloud platform.

### Co-immunoprecipitation (co-IP) analysis

For the exogenous co-IP assay, 293T cells were transfected with plasmids and incubated for 24 h, then lysed with IP lysis buffer (50 mM Tris–HCl, pH 7.4, 150 mM NaCl, 0.1% sodium deoxycholate, 0.1% [wt/vol] SDS, 0.1mM EDTA, 1% [vol/vol] NP-40, with 1 mM PMSF and 1 × phosphatase inhibitor added for fresh use) for 30 min at 4°C. After taking the input sample out, the rest were co-incubated with anti-FLAG (A2220; Sigma-Aldrich) or anti-HA beads (E6779; Sigma-Aldrich) for 2 h at 4°C to capture protein complexes. Otherwise, for endogenous co-IP assay, HeLa cells were infected with HAdV (MOI = 2) for 24 h and lysed using the same lysis buffer. Instead, the lysate was co-incubated with IP-grade anti-GSK3A antibodies (MA5-35239; Invitrogen) overnight at 4°C, after the addition of protein A/G agarose (sc-2003; Santa Cruz) for another 2 h of binding at 4°C.

After enrichment, the IP washing buffer (255 mM NaCl, 20 mM Tris–HCl, pH 7.4, 0.1 mM EDTA, 2.5 mM sodium deoxycholate, 0.2% [vol/vol] NP-40) was used to remove nonspecific combinations three to five times, each for 10 min at 4°C. Then, the beads were collected by centrifugation at 3,000 rpm for 1 min at 4°C. Both input and IP samples were denatured at 99°C with SDS–PAGE loading buffer for 10 min for further analysis.

### Protein structure prediction and protein–protein docking

The structure conformation of the viral L4-22K protein was modeled using a platform powered by ColabFold (based on AlphaFold v2.3.2) (Jumper et al, 2021). Computational docking of the predicted L4-22K structure with the established GSK3A structure (UniProt ID: P49840, PDB ID: 7SXF) was achieved using ClusPro 2.0 (cluspro.org) for protein–protein docking (Desta et al, 2020; Jones et al, 2022). Output clusters were visualized using PyMOL software (v3.1.1).

### Phylogenetic analysis

Viral L4-22K sequences from reference strains of various types of HAdV were retrieved from GenBank. These sequences were aligned, and a phylogenetic tree was constructed using the maximum-likelihood method with MEGA X software, with bootstrap values of 1,000 replicates.

### Statistics

Statistical analysis was conducted with GraphPad Prism 10 (v10.0.3). For normally distributed data, the *t* test was applied for the comparison of two groups, and one-way analysis of variance was for multiple groups. Each experiment was done in at least three biological replicates. The statistical significance was indicated as * when $P \leq 0.05$, ** when $P \leq 0.01$, *** when $P \leq 0.001$, **** when $P \leq 0.0001$.

# Data Availability

All data supporting the findings of this study are available within the article. The high-throughput screening and proteomics datasets contain information related to ongoing or unpublished projects and are therefore not publicly available. These datasets can be accessed from the corresponding author on reasonable request and upon completion of a material transfer agreement if required. Data will be made publicly available after the related follow-up studies are published or upon reasonable inquiry, in accordance with journal policy.

# Supplementary Information

# Acknowledgements

This work was supported by grants from the National Natural Science Foundation of China (32470141, 82072266); the CAMS Innovation Fund for Medical Sciences (2019-I2M-5-026); the National Key Research and Development Program of China (2023YFC2306001); the Beijing Municipal Administration of Hospitals Incubating Program (PX2024043); Beijing Research Center for Respiratory Infectious Diseases Project (BJRID2025-008); and Reform and Development of Beijing Municipal Health Commission.

## Author Contributions

Y Lin: resources, data curation, software, formal analysis, investigation, visualization, methodology, and writing—original draft, review, and editing.

Y Zhu: resources, supervision, funding acquisition, methodology, project administration, and writing—review and editing.

L Jing: resources, data curation, validation, and visualization.

Y Chen: resources, investigation, and methodology.

X Xiao: resources, data curation, supervision, investigation, and methodology.

X Lei: conceptualization, supervision, methodology, project administration, and writing—original draft, review, and editing.

Z Xie: conceptualization, resources, supervision, funding acquisition, project administration, and writing—review and editing.

## Conflict of Interest Statement

The authors declare that they have no conflict of interest.

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
