## [Reviewer comments · Life Science Alliance]

Life Science Alliance

GSK3A promotes human adenovirus replication and phosphorylates viral L4-22K protein

Ying Lin, Yun Zhu, Ling Jing, Yongjun Chen, Xia Xiao, Xiaobo Lei and Zheng de Xie

DOI: <https://doi.org/10.26508/lsa.202503320>

Corresponding author(s): Prof. Zheng de Xie (Beijing Children's Hospital)

Review Timeline:

Submission Date:	2025-03-21
Editorial Decision:	2025-04-21
Revision Received:	2025-05-13
Editorial Decision:	2025-06-06
Revision Received:	2025-06-08
Accepted:	2025-06-09

Scientific Editor: Tim Fessenden

Transaction Report:

April 21, 2025

Re: Life Science Alliance manuscript #LSA-2025-03320-T

Prof. Zheng de Xie
Beijing Children's Hospital, Capital Medical University, National Center for Children's Health
Laboratory of Infection and Virology
China

Dear Dr. Xie,

Thank you for submitting your manuscript entitled "GSK3A promotes human adenovirus replication by phosphorylating viral L4-22K protein" to Life Science Alliance. The manuscript was assessed by expert reviewers, whose comments are appended to this letter. We invite you to submit a revised manuscript addressing the Reviewer comments.

As you will see, both reviewers commended the intriguing results on adenoviruses obtained using an overexpression screen and phosphoproteomics. While reviewers diverged in their enthusiasm for these findings, both noted areas for improvement.

Namely, Reviewer 1 requested validation for increased DBP expression upon GSK3B overexpression and suggested immunofluorescence here. They raised concerns over discordant results shown in Fig 6 that should be addressed with new data or corrected quantifications. This reviewer also suggested new data to validate observations in Fig 2, however we concur with their alternative suggestion to tone down the conclusions related to this figure. Reviewer 2 felt the main conclusions of this study may be inaccurate given the reduced expression of the essential early viral gene E1A. A suitably revised manuscript should show if GSK3B binds to the E1A promoter, as requested, and discuss the significance of this potential repression. Finally, both reviewers felt that claims made on GSK3B interaction with L4-22K were not fully substantiated. The conclusions on this must be toned down in a revised manuscript (with the title change suggested by Reviewer 2), unless the phosphomutants are tested in a viral replication assay as sought by Reviewer 1.

Thank you for this interesting contribution to Life Science Alliance. We are looking forward to receiving your revised manuscript.

Sincerely,

- A letter addressing the reviewers' comments point by point.
- An editable version of the final text (.DOC or .DOCX) is needed for copyediting (no PDFs).
- High-resolution figure, supplementary figure and video files uploaded as individual files: See our detailed guidelines for

preparing your production-ready images, <https://www.life-science-alliance.org/authors>

B. MANUSCRIPT ORGANIZATION AND FORMATTING:

Reviewer #1 (Comments to the Authors (Required)):

In Lin et al., the authors performed a high-throughput gain-of-function screen by transfecting HeLa cells with a library of 15 000 genes to investigate their role in Human adenovirus (HAdV) infection. Through this approach, they identified the glycogen synthase kinase 3 α (GSK3A) as a proviral host factor for species B HAdV.

The first part of the study validates and characterizes this phenotype. The authors found that GSK3A expression promotes the expression of the viral genes at the RNA and protein levels. Conversely, GSK3A knockdown and knockout impair viral gene expression as well as viral production. GSK3A knockout does not affect viral entry but appears to impact viral replication at intermediate time points.

The second part of the publication uses phosphoproteomics to identify putative GSK3A targets and address the role of these phosphorylations for HAdV. Authors focus on the interaction between GSK3A and the viral protein L4-22K, and hypothesize a model in which GSK3A promotes L4-22K phosphorylation by interacting through GSK3A kinase domain. By doing so, GSK3A promotes L4-22K function and species B HAdV replication.

Overall, the study is well conducted and interesting. The authors performed two challenging high-throughput approaches by doing a gain-of-function screen and a phosphoproteomics survey. The knockout and knockdown experiments clearly demonstrate the importance of GSK3A for viral gene expression and HAdV infection in cell lines. The experimental design and execution meet the required standards for high-quality research.

Minor comments:

- Some conclusions on Western blots could be nuanced or elaborated:

o WB in Figure 1 has no quantification and is not very striking but is used in the discussion to exclude a potential role of GSK3B. Are there any other experiments involving GSK3B function? If not, maybe the discussion should temper the conclusions on GSK3B.

o Figure 2F, for HAdV other than HAdV-B3, DBP signal seems stronger. This is fine but this unique panel drive the conclusion than GSK3A specifically promotes species B HAdV. Are there any replicates and quantifications maybe? Do authors have other data related to this panel, qPCR or TCID50? If not, authors should maybe nuance their paragraph title and conclusion, and mention that GSK3A specificity remains to be further investigated.

o WB Figure 6A: I think there is some confusion in this figure, the quantification refers to IP:flag/WCL but no flag signal is shown. The bands used for quantification should be clearly stated in legend for example. The quantification and conclusions do not seem to reflect the WB shown.

o WB Figure 6C: a comment may be added in the main text to mention that the D2 mutant is not expressed. Similarly, a stronger interaction with the D4 VS D3 mutant is not obvious and no quantification is shown here.

- The importance of GSK3A kinase activity for HAdV infection is poorly discussed but somewhat important for concluding about the role of viral protein phosphorylations. Figure 5B and C show that rescuing cells with GSK3A inactive mutants still promotes viral production compared to the control. Viral productions are even close to the one with WT GSK3A. This suggest that GSK3A

function is partially dependent on its kinase activity. A comment could be added stating that GSK3A function is partially due to its kinase activity, but kinase-independent function cannot be excluded.

- GSK3A proviral function and its link to L4-22K could be more elaborated to provide a general model that connect the whole study. GSK3A seems to promote the transcription of several viral genes, do the authors think this is specific for certain early/late genes or all viral genes? Can authors discuss if GSK3A is more likely to impact early transcription mediated by host transcription factors or transcription mediated by viral proteins? Do authors know if L4-22K mRNA is affected by GSK3A knockout? If viral proteins synthesis is affected, do the loss of phosphorylations on viral proteins due to less kinase activity or less overall protein levels? If multiple viral proteins are affected by GSK3A knockout, is it likely to be mediated solely by L4-22K phosphorylation? How do point mutations in L4-22K (Figure 6D) would affect GSK3A binding if they are not in binding sites?

- As no data support the importance of L4-22K phosphorylation in viral infection or viral gene expression, I suggest changing the title to "GSK3A promotes human adenovirus replication and phosphorylates viral L4-22K protein"

I recommend the authors to address these minor concerns that mostly needs edits in the main text and discussion. I provide additional notes raising potential confusions in the manuscript. I recommend the authors to address them and state if they corrected the manuscript.

Notes:

METHODS:

- Not sure the primers names are correct for GSK3A: written "HAdV-B7 GSK3A forward" and "HAdV-B7 GSK3A" like it was a viral protein

- Co-immunoprecipitation (Co-IP) analysis: please specify the composition of the IP lysis buffer and wash buffer

- Plasmids: "pLenti-CRISPRv2, et al." ?

RESULTS:

Related to figure 1 :

-It is mentioned that the screen identified "identified 75 proviral host factors that promoted viral infections and 63 antiviral host factors". I am not sure that the red and blue dots in Fig1B reflects these numbers.

-The dose dependent effect of GSK3A expression on the viral protein DBP is not very striking. A picture of the immunofluorescence staining performed after the screen, showing the effect of GSK3A overexpression on DBP, could consolidate the blot.

Related to figure 2 :

-Figure 2E: looks like n=6 for the WT condition and n=7 in KO1. It is mentioned in the methods section that all experiments are n=3 but some of them have more than 3 values.

-Figure 2F is not described chronologically after 2E in the manuscript; figure 3 is described and then 2F.

-Figure 6D legend. I think is it RT-qPCR instead of qPCR

Related to figure 3 :

-Figure 3A: if may be worth specifying in the legend if the protein quantification measure only DBP levels or if it is normalized to the Actin signal

-Figure 3C " TCID50 assay further confirmed a notable increase in cells transfected with 500 ng GSK3A". An increase in viral production could be specified

-" Our findings suggested that GSK3A functions as a critical proviral host factor that enhances viral replication and transmission." I think it would be more adapted to talk about viral production rather than "transmission"

Related to figure 4 :

-Figure 4A: the main text states that the binding and internalization assays were performed with a MOI of 10, it is written 20 in the methods section.

-Figure 4B: no individual values shown in contrary to all other figures

Related to figure 5 :

-Figure 5B-C. The expression of GSK3A inactive mutants shows a decrease in DBP expression and viral production compared to cells expressing GSK3A, while no significance is shown in figure 5C. Is the effect of these mutants should be rather compared to control cells that don't express GSK3A? Is the increase in viral titers, compared to Vector condition, upon the expression of inactive mutants expected?

-Figure 5 legend. Some format inconsistencies: no bold format for A, B and C and no capital for "The workflow" or "The pie chart". I think G and H legends are wrong and it lacks J and K figure legends

Related to figure 6 :

-Figure 6A. The conclusions about the mutants are hard to notice. The S21A active mutant sounds decreasing L4-22k phosphorylation. The inactive mutants seem to bind more GSK3A. The quantification seems to go in the opposite trend of what can be seen on the blot. Has the IP/blot been repeated? No precision in legends. Maybe the quantification criteria should be specified in the legend. The interaction between GSK3A and L4-22K seems quantified using a flag-IP condition, there is no mention of flag-IP or staining on this blot. This figure and conclusions are a bit confusing and probably need to be more elaborated or nuanced.

-Figure 6C. Please specify that the D2 mutant is not stable or expressed. Quantification of several blots should maybe be added. It is difficult to see a better GSK3A binding to the D4 mutant vs D3.

-Figure 6D. The term "in vivo" is probably overstating considering they are IPs from transfected HEK293T cells

-Figure 6E. Please provide scores for protein structures and interactions from the Google ColabFold prediction

-Figure 6A legend. "Cells lysates were enriched using HA-tagged beads". Should it be "L4-22K-HA protein was enriched in cells lysates using HA-tagged beads"?

Reviewer #3 (Comments to the Authors (Required)):

Lin and colleagues report that glycogen synthase kinase-3alpha (GSK3A) promotes adenovirus (AdV) type B7 replication by phosphorylating the viral 22K protein. GSK3A is a dual specificity kinase, autophosphorylated on Ser / Thr residues, and its kinase activity is stabilized by tyrosine phosphorylation.

From an overexpression screen as well as mutant GSK3A protein expression in GSK3A KO HeLa cells, the authors suggest that the kinase activity of GSK3A affects the phosphorylation of S78/81 in the viral L4-22K protein, during the intermediate late phase of AdV-B7 infection. They claim that this is important for HAdV replication.

Further, the KO of GSK3A also led to the reduction of E1A and DBP phospho-serine levels suggesting some kind of indirect connection between GSK3A and the E1A and DBP proteins. Authors suggest that GSK3A interacts with L4-22K, E1A and DBP in co-immunoprecipitates. These immunocomplexes contain many different factors and hence conclusions from such experiments are weak. In such immunocomplexes, only L4-22K phosphorylation appeared to be dependent on catalytically active GSK3A.

Most importantly, however, the knock-down of GSK3A not only reduced intermediate and late viral gene expression, but very strongly blocked immediate early viral E1A expression, both at the mRNA and the protein levels. And this is a crucial result, since E1A is the master regulator in AdV infection.

Assessment:

The central message cannot be sustained, namely that 'GSK3A promotes human adenovirus replication by phosphorylating viral L4-22K protein (title)'. E1A is essential for the expression of all subviral promoters that give rise to protein coding transcripts. Rather, GSK3A seems to control the immediated early viral gene expression, but this effect is not studied here.

Furthermore, the study does not provide direct phosphorylation data with purified reactants to demonstrate that GSK3A and not some other kinase or regulatory factor of a phosphatase in cells, or in the immunoprecipitation complexes accounts for the phosphorylation events reported here.

Conclusion:

To substantiate their claim, the authors should mutate L4-22K S78 and S81 to constitutive negatively charged amino acids or nonphosphorylatable ones in the context of virus, and assess the replication levels of these viruses compared to control virus.

The reduction of GSK3A diminishes E1A transcript levels. The authors should test if GSK3A regulates the E1A promoter of B7, in a comparative analysis with the E1A promoter of C2/C5 viruses. Results could then be discussed in the context of available literature.

Other points:

- 1) Please provide the primary reference that AdV-B7 leads to toxic encephalopathy (see introduction).
- 2) Please provide the contextual information for how MIB1, IL8, TBK1 or USP7 enhance AdV infection. Which one of these factors affect the B-types that respond to GSK3A?
- 3) It is not clear how these cellular factors are supposed to be proviral. And how are the mentioned antiviral factors supposed to act against the virus?
- 4) The data description of Fig. 2F follows only after Fig. 3. Consider making an extra figure 4 with the data in Fig. 2F.

Point-by-point Response to Reviewer Comments

We sincerely thank the editor and all reviewers for their interest in our work and their valuable comments to improve the quality of our manuscript. Below we provide a point-by-point response (normal font) to each reviewer comment (italicized font). And all the corresponding changes to each comment were made in the revised manuscript and highlighted in dark red.

Reviewer #1 (Comments to the Authors (Required)):

In Lin et al., the authors performed a high-throughput gain-of-function screen by transfecting HeLa cells with a library of 15 000 genes to investigate their role in Human adenovirus (HAdV) infection. Through this approach, they identified the glycogen synthase kinase 3 α (GSK3A) as a proviral host factor for species B HAdV. The first part of the study validates and characterizes this phenotype. The authors found that GSK3A expression promotes the expression of the viral genes at the RNA and protein levels. Conversely, GSK3A knockdown and knockout impair viral gene expression as well as viral production. GSK3A knockout does not affect viral entry but appears to impact viral replication at intermediate time points. The second part of the publication uses phosphoproteomics to identify putative GSK3A targets and address the role of these phosphorylations for HAdV. Authors focus on the interaction between GSK3A and the viral protein L4-22K, and hypothesize a model in which GSK3A promotes L4-22K phosphorylation by interacting through GSK3A kinase domain. By doing so, GSK3A promotes L4-22K function and species B HAdV replication. Overall, the study is well conducted and interesting. The authors performed two challenging high-throughput approaches by doing a gain-of-function screen and a phosphoproteomics survey. The knockout and knockdown experiments clearly demonstrate the importance of GSK3A for viral gene expression and HAdV infection in cell lines. The experimental design and execution meet the required standards for high-quality research.

Re: Thank you for providing such helpful and precise suggestions for this manuscript. As your suggestion, we have carefully addressed all comments in the revised manuscript and corrected the notes you mentioned in the Methods and Results sections.

Minor comments:

*- Some conclusions on Western blots could be nuanced or elaborated:
o WB in Figure 1 has no quantification and is not very striking but is used in the discussion to exclude a potential role of GSK3B. Are there any other experiments involving GSK3B function? If not, maybe the discussion should temper the conclusions on GSK3B.*

Re: As suggested, we conducted GSK3B overexpression in WT and GSK3A-KO cells, respectively, and assessed its impact on HAdV replication. These results confirmed that GSK3B can not enhance viral replication. These findings have been incorporated into revised manuscript as Fig.1D, F and Fig.3G, H, I.

o Figure 2F, for HAdV other than HAdV-B3, DBP signal seems stronger. This is fine but this unique panel drive the conclusion than GSK3A specifically promotes species B HAdV. Are there any

replicates and quantifications maybe? Do authors have other data related to this panel, qPCR or TCID₅₀? If not, authors should maybe nuance their paragraph title and conclusion, and mention that GSK3A specificity remains to be further investigated.

Re: Thank you for your valuable suggestions. We conducted three independent replicates of this experiment, and the quantification for the WB analysis is presented in Fig. 4A and B. The DBP signal for HAdV-B3 was consistently weaker compared to other HAdV types. This difference may reflect inherent variations in viral replication kinetics or DBP expression timing among the different strains, despite using the same MOI and time point post-infection. Additionally, we have incorporated the RT-qPCR data of viral DBP transcript levels in Fig. 4C and the viral titers obtained from TCID₅₀ assays in Fig. 4D. Together, these results further support the conclusion that GSK3A seems to be a broad-spectrum proviral factor of HAdVs, with a particularly pronounced effect on Species B. Accordingly, we have revised the related content in the manuscript.

o WB Figure 6A: I think there is some confusion in this figure, the quantification refers to IP:flag/WCL but no flag signal is shown. The bands used for quantification should be clearly stated in legend for example. The quantification and conclusions do not seem to reflect the WB shown.

Re: Thank you for pointing out this mistake regarding Fig. 6A. We clarified the quantification details in the legend. Please refer to Fig. 7A in the revised manuscript.

The L4-22K-p band was observed by immunoblotting with the anti-pan-Ser/Thr-phosphorylation antibodies. The relative L4-22K-p level was calculated by the grey value of the 40kDa phosphorylation band divided by that of the anti-HA band in IP samples (the total protein level of HA-tagged L4-22K). As co-expressed with vector, wildtype GSK3A, S21A, K148A and Y279A mutants, L4-22K was enriched with HA beads and phosphorylation of HA-enriched L4-22K was detected. And the enriched L4-22K-p protein level was normalized by the grey value of HA-enriched L4-22K bands. Considering the phosphorylation signals of L4-22K (about 40 kDa) were not strong enough to convince the regulation, we then used HA beads. The quantification level of L4-22K-p regulated by GSK3A kinase activity when co-expressing with 3 mutants were normalized.

[Figure removed by editorial staff per authors' request]

o WB Figure 6C: a comment may be added in the main text to mention that the D2 mutant is not expressed. Similarly, a stronger interaction with the D4 VS D3 mutant is not obvious and no quantification is shown here.

Re: Thank you for your suggestion. To further investigate the functional domains of viral L4-22K protein, we generated new truncation mutants (D1, D2, D3, D4 and a combined D1-D4 mutant as indicated in Fig.7D). The finding indicated that only deletions of 92-168 aa disrupted the interaction between L4-22K and GSK3A, which is consistent with the protein-protein docking predictions. To quantify the binding affinities affected by truncations, we measured band intensities L4-22K truncation mutants and GSK3A protein by normalizing WT L4-22K and mutants IP signals to their input levels, and divided by the normalized GSK3A protein Co-IP signal. This method controls for differences in protein expression and IP efficiency, allowing comparison of binding affinities. Statistical analysis was performed on data from at least three independent experiments.

- The importance of GSK3A kinase activity for HAdV infection is poorly discussed but somewhat important for concluding about the role of viral protein phosphorylations. Figure 5B and C show that rescuing cells with GSK3A inactive mutants still promotes viral production compared to the control. Viral productions are even close to the one with WT GSK3A. This suggest that GSK3A function is partially dependent on its kinase activity. A comment could be added stating that GSK3A function is partially due to its kinase activity, but kinase-independent function cannot be excluded.

Re:

Thank you for your insightful comment. We agree on this and have added the comment to the discussion section.

- GSK3A proviral function and its link to L4-22K could be more elaborated to provide a general model that connect the whole study. GSK3A seems to promote the transcription of several viral genes, do the authors think this is specific for certain early/late genes or all viral genes? Can authors discuss if GSK3A is more likely to impact early transcription mediated by host transcription factors or transcription mediated by viral proteins? Do authors know if L4-22K mRNA is affected by GSK3A

knockout? If viral proteins synthesis is affected, do the loss of phosphorylations on viral proteins due to less kinase activity or less overall protein levels? If multiple viral proteins are affected by GSK3A knockout, is it likely to be mediated solely by L4-22K phosphorylation? How do point mutations in L4-22K (Figure 6D) would affect GSK3A binding if they are not in binding sites?

Re:

Thank you for asking these, very helpful and inspiring.

First, we are not sure about GSK3A's role in the spectrum of certain viral genes, but we assume it affects a wide range of viral genes. In our study, both early (E1A, DBP) and late (L4-22K, Hexon) gene transcripts were reduced by GSK3A KO, suggesting a potential broad transcriptional regulation. But this requires further investigation.

Second, considering that GSK3A is a pluripotent kinase, we believed that GSK3A has the chance to phosphorylate host transcription factors to adjust viral gene transcription, or phosphorylate important viral proteins like L4-22K, affecting virus gene transcription. We detected the restriction effects of GSK3A depletion on viral L4-22K transcriptions below and found that depletion of GSK3A broadly restricts HAdV-B~E strains, except for C1. Since viral L4-22K was found to regulate virus transcriptions, the role of GSK3A KO on E1A, DBP, or Hexon transcription might be related to L4-22K functions that are affected by phosphorylation. So we believe it is important to uncover the function of phosphorylation of L4-22K. However, we failed to investigate the role of GSK3A on L4-22K protein levels due to the lack of specific antibodies. So, it is unclear whether it regulated L4-22K protein synthesis. We assumed that phosphorylation of L4-22K regulated by GSK3A may affect L4-22K total protein level, or its activity in stimulating virus transcription, viral RNA transportation, and so on. To figure this out, there required more investigations. And we have added relative statements in the discussion sections as you suggested.

[Figure removed by editorial staff per authors' request]

Re:

Thank you for the advice, we have changed the title as you recommended. We also acknowledge that the role of L4-22K phosphorylation in viral infection and gene expression remains to be elucidated, and we plan to investigate this important aspect further in future studies.

I recommend the authors to address these minor concerns that mostly needs edits in the main text and discussion. I provide additional notes raising potential confusions in the manuscript. I recommend the authors to address them and state if they corrected the manuscript.

Re:

Thank you for being patient and giving constructive feedback. We have carefully addressed the minor concerns and potential points of confusion you raised, making corresponding edits in the main text and discussion sections to enhance clarity and precision.

Notes:

METHODS:

- Not sure the primers names are correct for GSK3A: written "HAdV-B7 GSK3A forward" and "HAdV-B7 GSK3A" like it was a viral protein

Re:

Sorry about this, we have corrected the primer names in the revised manuscript, removing the "HAdV-B7" prefix to avoid misunderstanding.

- Co-immunoprecipitation (Co-IP) analysis: please specify the composition of the IP lysis buffer and wash buffer

Re:

Sorry about this, we have added the composition information of the IP lysis buffer and washing buffer in the method section, to ensure reproducibility and clarity for readers.

- Plasmids: "pLenti-CRISPRv2, et al." ?

Re:

Sorry about this, we have corrected this in the revised manuscript.

RESULTS:

Related to figure 1 :

-It is mentioned that the screen identified "identified 75 proviral host factors that promoted viral infections and 63 antiviral host factors". I am not sure that the red and blue dots in Fig1B reflects these numbers.

Re:

Sorry, we have corrected this in Fig.1B.

-The dose dependent effect of GSK3A expression on the viral protein DBP is not very striking. A picture of the immunofluorescence staining performed after the screen, showing the effect of GSK3A overexpression on DBP, could consolidate the blot.

Re:

Thank you, we have added the IF results and quantification data as regarded to Fig.1D.

Related to figure 2 :

-Figure 2E: looks like n=6 for the WT condition and n=7 in KO1. It is mentioned in the methods section that all experiments are n=3 but some of them have more than 3 values.

Re:

Thank you for mentioning this. We have checked the raw data, and in fact, it's n=7 in both groups, but we found that 1 dot in the WT group was hidden by the others. The TCID₅₀ values in the two groups are shown below. We have corrected the statement of sample values in the methods section and the figure legend of Fig. 2.

[Figure removed by editorial staff per authors' request]

-Figure 2F is not described chronologically after 2E in the manuscript; figure 3 is described and then 2F.

Re:

We have corrected this in order, turning Fig. 2F into Fig. 4, following Fig. 3.

-Figure 6D legend. I think is it RT-qPCR instead of qPCR

Re:

Sorry about this, we suppose you might be referring to Figure 2D instead of Figure 6D. We have updated the figure legend of Fig. 2D to replace “qPCR” with “RT-qPCR” accordingly.

Related to figure 3 :

-Figure 3A: if may be worth specifying in the legend if the protein quantification measure only DBP levels or if it is normalized to the Actin signal

Re:

We apologize for the omission; the quantification measure of DBP protein in Fig. 3A was normalized to the actin signal. We have added this clarification to the figure legend of Fig. 3A.

Figure 3C " TCID50 assay further confirmed a notable increase in cells transfected with 500 ng GSK3A". An increase in viral production could be specified

Re:

Thank you for your insightful comment, we appreciate your suggestion to specify the viral production increase. We have modified this in the results section.

-" Our findings suggested that GSK3A functions as a critical proviral host factor that enhances viral replication and transmission." I think it would be more adapted to talk about viral production rather than "transmission"

Re:

Thank you, we have corrected this in the results section by replacing “transmission” with “production”.

Related to figure 4 :

-Figure 4A: the main text states that the binding and internalization assays were performed with a MOI of 10, it is written 20 in the methods section.

Re:

Sorry for the mistake, we have corrected this in the methods section.

-Figure 4B: no individual values shown in contrary to all other figures

Re: We apologize for this inconsistency and have revised the analysis with individual values shown. The result has been reorganized into Fig.5B in the revised manuscript.

Related to figure 5 :

-Figure 5B-C. The expression of GSK3A inactive mutants shows a decrease in DBP expression and viral production compared to cells expressing GSK3A, while no significance is shown in figure 5C. Is the effect of these mutants should be rather compared to control cells that don't express GSK3A? Is the increase in viral titers, compared to Vector condition, upon the expression of inactive mutants expected?

Re:

We apologize for any lack of clarity in our previous description. Our experimental design involved two key comparisons, 1) wild-type GSK3A versus empty vector controls, and 2) all mutants versus WT GSK3A. The results demonstrate that WT GSK3A significantly enhanced viral replication compared to empty vector controls. Among the mutants, the constitutively active S21A mutant further increased viral replication compared to WT GSK3A, while the phosphorylation-deficient mutants Y297A and K148A markedly attenuated GSK3A's proviral activity. Although cells transfected with kinase-dead mutants (Y279A/K148A) exhibited moderate but statistically significant increases in

viral replication compared to vector controls, this enhancement remained significantly weaker than that mediated by wild-type GSK3A.

The partial rescue by inactive mutants suggests that GSK3A may also exert proviral effects through non-enzymatic mechanisms. For example, GSK3A may stabilize L4-22K or recruit other transcriptional cofactors based on physical interactions. This interpretation has been added to the Discussion section in the revised manuscript. We appreciate your valuable feedback, which has strengthened the clarity and rigor of our discussion.

-Figure 5 legend. Some format inconsistencies: no bold format for A, B and C and no capital for "The workflow" or "The pie chart". I think G and H legends are wrong and it lacks J and K figure legends

Re:

Thank you for pointing these out. We have corrected the formatting issues in the revised manuscript. We apologize for the oversight and have added and corrected the figure legends for panels G~K accordingly.

Related to figure 6 :

-Figure 6A. The conclusions about the mutants are hard to notice. The S21A active mutant sounds decreasing L4-22k phosphorylation. The inactive mutants seem to bind more GSK3A. The quantification seems to go in the opposite trend of what can be seen on the blot. Has the IP/blot been repeated? No precision in legends. Maybe the quantification criteria should be specified in the legend. The interaction between GSK3A and L4-22K seems quantified using a flag-IP condition, there is no mention of flag-IP or staining on this blot. This figure and conclusions are a bit confusing and probably need to be more elaborated or nuanced.

Re:

Sorry about the lack of precision in the legends. We repeated the IP/blot and recalculated this, and the results were shown before in the third reply to your minor comments. We also replaced the quantitative data in Figure 7A (primarily Figure 6A) and added the corresponding information in the figure legend.

-Figure 6C. Please specify that the D2 mutant is not stable or expressed. Quantification of several blots should maybe be added. It is difficult to see a better GSK3A binding to the D4 mutant vs D3.

Re:

Thank you for your valuable comments. To improve clarity, we generated new truncations of the viral L4-22K protein based on the protein-protein docking results, including D1~D4 and the total D1~4 mutant. The expression of all the truncated plasmids was confirmed. The new results were consistent with the docking analysis and are shown in Fig. 7D in the revised manuscript, replacing the former ones. We have also made corresponding changes in the results and discussion sections.

-Figure 6D. The term "in vivo" is probably overstating considering they are IPs from transfected HEK293T cells

Re: We acknowledge the inappropriateness of using “*in vivo*”. We have therefore removed it in the revised manuscript.

-Figure 6E. Please provide scores for protein structures and interactions from the Google ColabFold prediction.

Re:

For L4-22K predicted structure, the sequence coverage data is indicated here:

[Figure removed by editorial staff per authors' request]

Here is the structure colored by pLDDT generated with Chimera-X with default color.

[Figure removed by editorial staff per authors' request]

This is the predicted aligned errors (PAE) data generated with Chimera-X with default color. Blue indicates low PAE values and high confidence area. Red indicated high PAE values with low confidence regions.

[Figure removed by editorial staff per authors' request]

For protein-protein docking, the cluspro scores of the outputting 30 GSK3A-L4-22K complex models were shown below:

Cluster	Members	Representative	Weighted Score
0	48	Center	-1286.1
0	48	Lowest Energy	-1410.5

We choose this model due to the highest stability among the 48 members in this cluster.

-Figure 6A legend. "Cells lysates were enriched using HA-tagged beads". Should it be "L4-22K-HA protein was enriched in cells lysates using HA-tagged beads"?

Re: Sorry about this, we have corrected this in the Fig. 7A legend (primarily be Fig. 6A).

Reviewer #3 (Comments to the Authors (Required)):

Lin and colleagues report that glycogen synthase kinase-3alpha (GSK3A) promotes adenovirus (AdV) type B7 replication by phosphorylating the viral 22K protein. GSK3A is a dual specificity kinase, autophosphorylated on Ser / Thr residues, and its kinase activity is stabilized by tyrosine phosphorylation. From an overexpression screen as well as mutant GSK3A protein expression in GSK3A KO HeLa cells, the authors suggest that the kinase activity of GSK3A affects the phosphorylation of S78/81 in the viral L4-22K protein, during the intermediate late phase of AdV-B7 infection. They claim that this is important for HAdV replication. Further, the KO of GSK3A also led to the reduction of E1A and DBP phospho-serine levels suggesting some kind of indirect connection between GSK3A and the E1A and DBP proteins. Authors suggest that GSK3A interacts with L4-22K, E1A and DBP in co-immunoprecipitates. These immunocomplexes contain many different factors and hence conclusions from such experiments are weak. In such immunocomplexes, only L4-22K phosphorylation appeared to be dependent on catalytically active GSK3A.

Most importantly, however, the knock-down of GSK3A not only reduced intermediate and late viral gene expression, but very strongly blocked immediate early viral E1A expression, both at the mRNA and the protein levels. And this is a crucial result, since E1A is the master regulator in AdV infection.

Assessment:

The central message cannot be sustained, namely that 'GSK3A promotes human adenovirus replication by phosphorylating viral L4-22K protein (title)'. E1A is essential for the expression of all

subviral promoters that give rise to protein coding transcripts. Rather, GSK3A seems to control the immediated early viral gene expression, but this effect is not studied here.

Re:

Thank you for your comments. We investigated the effects of GSK3A depletion on viral E1A transcripts using RT-qPCR analysis which is shown below. The results indicated that GSK3A is essential for the E1A transcription of HAdV-B7, B11, C2, C5, and E4 strains. Therefore, we do not deny the potential control of GSK3A on E1A. But the co-IP analysis of GSK3A interacting with E1A protein was relatively weak, and we failed to detect the E1A phosphorylation band through IP enrichment as well. So, the mechanism of GSK3A regulating E1A expression was unclear based on the data we investigated so far. We will continue to study the potential effect and mechanism of GSK3A on E1A and other viral proteins of HAdV in our future studies.

[Figure removed by editorial staff per authors' request]

Furthermore, the study does not provide direct phosphorylation data with purified reactants to demonstrate that GSK3A and not some other kinase or regulatory factor of a phosphatase in cells, or in the immunoprecipitation complexes accounts for the phosphorylation events reported here.

Re:

Thank you for mentioning this. We have tried to purify the viral proteins and perform the in vitro phosphorylation experiments to provide direct data, but we have not achieved ideal production so far. We will make our efforts on this continuously in future studies. And a statement specifying the possibility of other kinases or regulatory factors co-stimulating viral protein phosphorylation was added in the revised manuscript in the discussion sections.

Conclusion:

To substantiate their claim, the authors should mutate L4-22K S78 and S81 to constitutive negatively charged amino acids or nonphosphorylatable ones in the context of virus, and assess the replication levels of these viruses compared to control virus.

Re:

We appreciate the reviewer's interest in our investigation of the functional role of L4-22K phosphorylation in HAdV-7 replication. In response to the reviewer's comments, we clarify that we are currently constructing an HAdV-7 infectious clone to serve as a platform for introducing phosphorylation-deficient point mutations S78A and S81A into the L4-22K gene. The system will enable us to rescue mutant viruses and compare their replication kinetics with wild-type HAdV-B7 to assess the impact of S78 and S81 phosphorylation on viral growth. However, we acknowledge the

technical challenges of manipulating the large adenovirus genome and are committed to addressing these experiments in future studies.

The reduction of GSK3A diminishes E1A transcript levels. The authors should test if GSK3A regulates the E1A promoter of B7, in a comparative analysis with the E1A promoter of C2/C5 viruses. Results could then be discussed in the context of available literature.

Re:

Thank you for your advice. We have added the analysis of GSK3A on the E1A transcript level in formal replies according to Tim's e-mail. Meanwhile, we made a phylogenetic analysis of E1A gene across HAdV-B3, B7, B11, E4, C2, C5 and D8. Since we have not found supportive literature, we are glad to investigate GSK3A's regulatory role on the E1A promoter in our future studies.

[Figure removed by editorial staff per authors' request]

Other points:

1) Please provide the primary reference that AdV-B7 leads to toxic encephalopathy (see introduction).

Re:

Thank you for your attention. We have included the relevant reference in the introduction section. The study by Fu et al., titled "Human adenovirus type 7 infection causes a more severe disease than type 3" (BMC Infectious Diseases, 2019, DOI: 10.1186/s12879-018-3651-2), reported that from 2009-2015, only one child infected with HAdV-B3 (n = 127) developed toxic encephalopathy, whereas 25 cases occurred secondary to HAdV-B7 infections (n = 131). This underscores the greater severity of HAdV-B7 infection than HAdV-B3 in clinical outcomes.

2) Please provide the contextual information for how MIB1, IL8, TBK1 or USP7 enhance AdV infection. Which one of these factors affect the B-types that respond to GSK3A?

Re:

We have added the contextual information on the details of the mechanisms of these factors enhancing HAdV infection. Unfortunately, none of them were found to affect HAdV-B Species like GSK3A do.

3) It is not clear how these cellular factors are supposed to be proviral. And how are the mentioned antiviral factors supposed to act against the virus?

Re:

We have added the details of the mechanisms of these factors promoting or restricting HAdV infection, as you suggested.

4) The data description of Fig. 2F follows only after Fig. 3. Consider making an extra figure 4 with the data in Fig. 2F.

Re:

Thank you for pointing this out, we have moved the data in Fig. 2F to Fig. 4 as you suggested.

We hope the revisions have addressed all concerns raised. Thank you again for your constructive feedback, which has greatly improved the quality of the manuscript. Please let us know if further clarifications are needed.

Sincerely,
Zhengde Xie, MD.

Beijing Key Laboratory of Pediatric Respiratory Infectious Diseases, Key Laboratory of Major Diseases in Children, Ministry of Education, National Clinical Research Center for Respiratory Diseases, Laboratory of Infection and Virology, Beijing Pediatric Research Institute, Beijing Children's Hospital, Capital Medical University, National Center for Children's Health
Nanlishi Road No. 56, Xicheng District, Beijing 100045, China
Tel: +86-10-13370110098
E-mail: xiezhendge@bch.com.cn

June 6, 2025

RE: Life Science Alliance Manuscript #LSA-2025-03320-TR

Prof. Zheng de Xie
Beijing Children's Hospital
Laboratory of Infection and Virology
Nanlishi Road No. 56, Xicheng District
Beijing 100045
China

Dear Dr. Xie,

Thank you for submitting your revised manuscript entitled "GSK3A promotes human adenovirus replication and phosphorylates viral L4-22K protein". We returned this work to Reviewer 1, whose comments are appended below. Please address the remaining minor requests from this reviewer, including improved description of the quantification of blots shown in Fig 7A and 7B. We would be happy to publish your paper in Life Science Alliance pending these changes and final revisions necessary to meet our formatting guidelines.

- Please add ORCID ID for secondary corresponding author--they should have received instructions on how to do so.
- Please add the X and Bluesky handles of your host institute/organization as well as your own or/and one of the authors in our system.
- Please consult our manuscript preparation guidelines <https://www.life-science-alliance.org/manuscript-prep> and make sure your manuscript sections are in the correct order.
- Please move your main legends in the main manuscript text after the references section.
- Please add an Author Contributions section to your main manuscript text.
- There are call-outs for supplementary figures in the manuscript text, but they have not been provided. Please correct.
- Please add call-outs for Figure 6D and J to your main manuscript text
- Please add scale bars to the images in Figure 1D.
- Please ensure that all protein are displayed in the highest resolution. The blots in Figure 1F do not match the associated source data, possibly due to low resolution.
- Please add a statement on institutional approval for use of human specimens.
- Please provide an institutional email address for co-corresponding author Xiaobo Lei.

A. FINAL FILES:

-- Summary blurb (enter in submission system): A short text summarizing in a single sentence the study (max. 200 characters including spaces). This text is used in conjunction with the titles of papers, hence should be informative and complementary to

the title. It should describe the context and significance of the findings for a general readership; it should be written in the present tense and refer to the work in the third person. Author names should not be mentioned.

B. MANUSCRIPT ORGANIZATION AND FORMATTING:

Sincerely,

Reviewer #1 (Comments to the Authors (Required)):

This manuscript, revised by Lin et al., addresses most of my comments and provides substantial edits by adding new data and revising some overstatements. Only some confusion remains regarding Figure 7, which should be addressed. The discussion has been revised to include data on GSK3B, an elaborated hypothesis about the role of L4-22K during infection, and the impact of GSK3A expression on different HAdV strains.

I recommend the authors to address the confusion related to figure 7 as discussed below.

-Figure 1 data support the absence of proviral function for GSK3B, thank you for these new figures

-New data in Figure 4 reinforce the conclusions, thank you. For some reasons, WB in the Figure 4A are a bit blurry on my monitor. The main text concludes "Together, these findings suggested that GSK3A is a crucial host factor that facilitates HAdV replication, with a specific role in the replication of HAdV species.". Do you mean HAdV B species or other specific species, or just HAdV?

-Regarding Figure 7A: A description of the quantification has been added, thank you. However, the quantification is hard to understand: "It was calculated by normalizing the IP-enriched bands of each mutant to their input levels, then divided by a normalized L4-22K IP signal, and standardized to the normalized binding affinity of WT GSK3A." I am Okay with that as it seems to match the blot, but it could be nice to improve this explanation. "standardized to the normalized binding affinity of WT GSK3A" but the vector condition is at 1. Is it normalized at 5 for WT GSK3A of normalized with the vector condition?

Optionally, you could add bar charts showing the normalized values of L4-22K IP , then the normalized pulled down of GSK3A, along with your quantification. This is OK, just a suggestion if you want to clarify this figure.

-The WB quantification figure shows the "L4-22K-p/L4-22K (normalized)" signal or there an additional normalization with the "the IP-enriched bands of each mutant to their input levels" ? Is there another quantification missing ? The main text is confusing as Fig. 7A has a WB, a "L4-22K-p/L4-22K (normalized)" signal plot and a "normalized binding index"? The normalized binding index is the L4-22K-p figure or something is missing? Maybe it needs to be referred as Fig. 7B for clarity.

Typos:

Line 561-562: DNA polymerase, et al?

Point-by-Point Response to Reviewer Comments

We sincerely thank the editor and the reviewer for their positive feedback to improve the quality of our manuscript. Below, we provide a point-by-point response (in normal font) to each comment (in italicized font). And all the corresponding changes to each comment were made in the revised manuscript and highlighted in dark red.

-Please add ORCID ID for secondary corresponding author--they should have received instructions on how to do so.

Reply:

The ORCID ID for the secondary corresponding author has been added.

-Please add the X and Bluesky handles of your host institute/organization as well as your own or/and one of the authors in our system.

Reply:

Thank you for your message. Unfortunately, due to restrictions in China, we are unable to access X or Bluesky and therefore do not have the accounts for our host institute or organization.

-Please consult our manuscript preparation guidelines <https://www.life-science-alliance.org/manuscript-prep> and make sure your manuscript sections are in the correct order.

Reply:

We have adjusted the orders of each section according to the guidelines.

-Please move your main legends in the main manuscript text after the references section.

Reply:

We have adjusted the orders of each section according to the guidelines, including moving the figure legends text after the reference section.

-Please add an Author Contributions section to your main manuscript text.

Reply:

We have adjusted the orders of each section according to the guidelines, including adding the summary blurb, author contribution and data availability statement.

-There are call-outs for supplementary figures in the manuscript text, but they have not been provided. Please correct.

Reply:

Sorry about this, we have deleted this description.

-Please add call-outs for Figure 6D and J to your main manuscript text

Reply:

Sorry, we have corrected this in the main text.

-Please add scale bars to the images in Figure 1D.

Reply:

Sorry, we have added the scale bars to Figure 1D.

-Please ensure that all proteins are displayed in the highest resolution. The blots in Figure 1F do not match the associated source data, possibly due to low resolution.

Reply:

Sorry, we have replaced Figure 1F with the highest resolution.

-Please add a statement on institutional approval for use of human specimens.

Reply:

We have added this statement in the method and materials section.

-Please provide an institutional email address for co-corresponding author Xiaobo Lei.

Reply:

We have added the email address for Xiaobo Lei.

Reviewer #1 (Comments to the Authors (Required))

This manuscript, revised by Lin et al., addresses most of my comments and provides substantial edits by adding new data and revising some overstatements. Only some confusion remains regarding Figure 7, which should be addressed. The discussion has been revised to include data on GSK3B, an elaborated hypothesis about the role of L4-22K during infection, and the impact of GSK3A expression on different HAdV strains. I recommend the authors to address the confusion related to figure 7 as discussed below.

-Figure 1 data support the absence of proviral function for GSK3B, thank you for these new figures

Reply:

Thank you for your acknowledgement of the new figures, which further strengthen our main conclusions.

-New data in Figure 4 reinforce the conclusions, thank you. For some reasons, WB in the Figure 4A are a bit blurry on my monitor. The main text concludes "Together, these findings suggested that GSK3A is a crucial host factor that facilitates HAdV replication, with a specific role in the replication of HAdV species.". Do you mean HAdV B species or other specific species, or just HAdV

Reply:

Sorry for the quality of WB results in Fig. 4A, we have replaced it with a new version of file.

According to our results, GSK3A KO significantly restricts B~E HAdV replication at the DBP transcription level. For DBP protein levels, most of the HAdV strains were significantly inhibited by GSK3A KO, especially Species B and E (with the lowest expression level of DBP in GSK3A-KO cells compared to WT cells), but several stains showed no statistical significance but also a down-regulatory trend (such as B11, C5, C6). For viral titers, most of the HAdV strains were significantly inhibited by GSK3A KO except for B14, C2, C6, and E4. The difference may relate to different virus life cycle phase lengths across the stains, or due to the different efficacy of anti-DBP antibodies across the strains. But from the whole perspective, combined with the RT-qPCR results and the microscopic observation of the viral lesions while we collect the cell samples (not showed in the manuscript), GSK3A seemed to a pan-proviral host factor for respiratory HAdVs including Species B~E.

-Regarding Figure 7A: A description of the quantification has been added, thank you. However, the quantification is hard to understand: "It was calculated by normalizing the IP-enriched bands of each mutant to their input levels, then divided by a normalized L4-22K IP signal, and standardized to the normalized binding affinity of WT GSK3A." I am Okay with that as it seems to match the blot, but it could be nice to improve this explanation. "Standardized to the normalized binding affinity of WT GSK3A" but the vector condition is at 1. Is it normalized at 5 for WT GSK3A of normalized with the vector condition? Optionally, you could add bar charts showing the normalized values of L4-22K IP, then the normalized pull-down of GSK3A, along with your quantification. This is OK, just a suggestion if you want to clarify this figure.

Reply:

We apologize for any confusion caused by the previous description. In the revised manuscript, we have rewritten this for greater clarity. Briefly, the quantification was performed as follows: (1) The IP-enriched bands of each mutant were first normalized to their respective input levels to account for loading differences; (2) these values were then divided by the normalized L4-22K IP signal to control for variations in immunoprecipitation efficiency; (3) finally, all binding affinities were standardized such that the vector control was set to 1, which resulted in the WT GSK3A condition being normalized to 5. We have clarified this calculation process in the figure legend.

Thank you very much for your helpful suggestion. While we believe the western blot in Figure 7A already illustrates the promoting role of GSK3A on L4-22K-p, we also provided quantification to

further support this observation. We agree that a clear explanation of the binding affinity calculation process is important and have revised our description accordingly. So we added the analysis on the column chart of binding affinity in Fig.7A. But the normalized L4-22K pull down levels were not included due to the space limitation. Thank you again for your valuable feedback and for helping us improve the clarity of our data presentation.

-The WB quantification figure shows the "L4-22K-p/L4-22K (normalized)" signal or there an additional normalization with the "the IP-enriched bands of each mutant to their input levels" ? Is there another quantification missing? The main text is confusing as Fig. 7A has a WB, a "L4-22K-p/L4-22K (normalized)" signal plot and a "normalized binding index"? The normalized binding index is the L4-22K-p figure or something is missing? Maybe it needs to be referred as Fig. 7B for clarity.

Reply:

Thank you for your thoughtful feedback and for pointing out the need for clarification. We apologize for any confusion we may have caused. In response, we have added the binding affinity analysis to Fig. 7A and updated the corresponding description in the revised manuscript. To summarize, we used two indices with different calculation methods. First, the normalized IP-enriched bands were included to illustrate the potential impact of GSK3A mutants on their binding affinity with L4-22K. Second, the "L4-22K-p/L4-22K" ratio directly shows the effect of GSK3A on L4-22K-p levels, normalized by the vector condition. We appreciate your suggestion and have clarified this rationale in the revised manuscript.

Typos:

Line 561-562: DNA polymerase, et al?

Sorry for the typing issue, we have corrected this in the manuscript.

June 9, 2025

RE: Life Science Alliance Manuscript #LSA-2025-03320-TRR

Prof. Zheng de Xie
Beijing Children's Hospital
Laboratory of Infection and Virology
Nanlishi Road No. 56, Xicheng District
Beijing 100045
China

Dear Dr. Xie,

Thank you for submitting your Research Article entitled "GSK3A promotes human adenovirus replication and phosphorylates viral L4-22K protein". It is a pleasure to let you know that your manuscript is now accepted for publication in Life Science Alliance. Congratulations on this interesting work.

DISTRIBUTION OF MATERIALS:

Again, congratulations on a very nice paper. I hope you found the review process to be constructive and are pleased with how the manuscript was handled editorially. We look forward to future exciting submissions from your lab.

Sincerely,
